# Missing self triggers NK cell-mediated chronic vascular rejection of solid organ transplants

Alice Koenig[1,2,3], Chien-Chia Chen [1], Antoine Marçais[1], Thomas Barba[1,2,3], Virginie Mathias[1,4], Antoine Sicard[1,2,3], Maud Rabeyrin[5], Maud Racapé[6], Jean-Paul Duong-Van-Huyen[6], Patrick Bruneval[6], Alexandre Loupy[6], Sébastien Dussurgey[7], Stéphanie Ducreux[1,4], Vannary Meas-Yedid[8], Jean-Christophe Olivo-Marin[8], Héléna Paidassi[1], Romain Guillemain[9], Jean-Luc Taupin[10,11,12], Jasper Callemeyn[13,14], Emmanuel Morelon[1,2,3], Antonino Nicoletti[12,15], Béatrice Charreau [16], Valérie Dubois[1,4], Maarten Naesens[13,14], Thierry Walzer[1,17], Thierry Defrance[1,17] & Olivier Thaunat[1,2,3]*

Current doctrine is that microvascular inflammation (MVI) triggered by a transplant -recipient antibody response against alloantigens (antibody-mediated rejection) is the main cause of graft failure. Here, we show that histological lesions are not mediated by antibodies in approximately half the participants in a cohort of 129 renal recipients with MVI on graft biopsy. Genetic analysis of these patients shows a higher prevalence of mismatches between donor HLA I and recipient inhibitory killer cell immunoglobulin-like receptors (KIRs). Human in vitro models and transplantation of β2-microglobulin-deficient hearts into wild-type mice demonstrates that the inability of graft endothelial cells to provide HLA I-mediated inhibitory signals to recipient circulating NK cells triggers their activation, which in turn promotes endothelial damage. Missing self-induced NK cell activation is mTORC1-dependent and the mTOR inhibitor rapamycin can prevent the development of this type of chronic vascular rejection.

[1] CIRI, INSERM U1111, Université Claude Bernard Lyon I, CNRS UMR5308, Ecole Normale Supérieure de Lyon, Univ. Lyon, 21, avenue Tony Garnier, 69007 Lyon, France. [2] Hospices Civils de Lyon, Edouard Herriot Hospital, Department of Transplantation, Nephrology and Clinical Immunology, 5, place d'Arsonval, 69003 Lyon, France. [3] Lyon-Est Medical Faculty, Claude Bernard University (Lyon 1), 8, avenue Rockfeller, 69373 Lyon, France. [4] French National Blood Service (EFS), HLA Laboratory, 111, rue Elisée-Reclus, 69153 Décines-Charpieu, France. [5] Hospices Civils de Lyon, Department of Pathology, 59, boulevard Pinel, 69500 Bron, France. [6] Paris Translational Research Centre for Organ Transplantation, Paris Descartes University, 12, rue de l'Ecole de Médecine, 75006 Paris, France. [7] SFR Biosciences (UMS3444/CNRS, US8/Inserm, ENS de Lyon, UCBL), 50, avenue Tony-Garnier, 69007 Lyon, France. [8] Unité d'Analyse d'Images Biologiques, Pasteur Institut, 25-28, rue du Docteur-Roux, 75015 Paris, France. [9] Assistance Publique - Hôpitaux de Paris, Georges Pompidou Hospital, Cardiology and Heart Transplant Department, 20, rue Leblanc, 75015 Paris, France. [10] Assistance Publique - Hôpitaux de Paris, Immunology and HLA Laboratory, Saint-Louis Hospital, 1, avenue Claude-Vellefaux, 75010 Paris, France. [11] French National Institute of Health and Medical Research (Inserm) Unit 1160, 1, avenue Claude-Vellefaux, 75010 Paris, France. [12] Paris Diderot University, 5, rue Thomas-Mann, 75013 Paris, France. [13] Department of Microbiology and Immunology, KU Leuven, University of Leuven, Herestraat 49, Box 7003, 3000 Leuven, Belgium. [14] Department of Nephrology and Renal Transplantation, University Hospitals Leuven, Herestraat 49, 3000 Leuven, Belgium. [15] French National Institute of Health and Medical Research (Inserm) Unit 1148, Laboratory of Vascular Translational Science, 46, rue Henri-Huchard, 75018 Paris, France. [16] French National Institute of Health and Medical Research (Inserm) UMR1064, 30, boulevard Jean-Monnet, 44093 Nantes Cedex 01, France. [17]These authors contributed equally: Thierry Walzer, Thierry Defrance. *email: olivier.thaunat@inserm.fr

The best (and often the only) therapeutic option for patients with end-stage vital organ failure is organ transplantation. However, the antigenic determinants that differ between the donor and the recipient (alloantigens), in particular the highly polymorphic molecules from the major histocompatibility complex (MHC, i.e. human leucocyte antigen (HLA) in humans), are recognised by the adaptive immune system of the recipient[1], which can lead to the failure of the transplanted organ, a process named "rejection".

Until the end of the 1970s, the occurrence of acute cellular rejection episodes, i.e. the destruction of the graft by the recipient's cytotoxic T cells, was the main obstacle to the success of transplantation. Introduction of calcineurin inhibitors in the early 1980s led to a dramatic reduction of the incidence of acute cellular rejection, and doubled the percentage of functional renal grafts at 1-year post-transplantation[2]. However, this progress in the control of T cell alloimmune response barely affected graft half-life[3], leading to the emergence of the "humoral theory" of chronic rejection[4].

First identified in renal transplantation in the 2000s[5–8], antibody-mediated rejection (AMR) has since been recognised as the main cause of failure in most organ transplantations[9–12].

Graft endothelium is the biological interface between donor alloantigens and host antibodies, which are retained in the recipient's circulation due to their size[13]. Binding of circulating anti-HLA donor-specific antibodies (DSAs) to directly accessible targets expressed by endothelial cells (ECs) of graft microvasculature can activate the classical complement pathway, thereby accelerating the rejection process[14,15]. This is, however, not mandatory for the development of chronic AMR[16,17]. Engagement of the surface Fc receptors of innate immune effectors by anti-HLA DSA bound to graft microvasculature is sufficient to trigger the release of lytic enzymes that mediate the activation and/or damage of the endothelial cell layer. For this reason, the presence of microvascular inflammation (MVI) in graft biopsy is widely considered as the histological hallmark of AMR[18,19].

Our present translational study challenges this idea. Analysing a cohort of 129 renal transplant patients we find that MVI in graft biopsy is not mediated by antibodies in almost half of the cases. Instead, genetic analyses suggest that microvascular lesions are a result of direct activation of recipient NK cells by graft ECs. We find that graft ECs are unable to deliver inhibitory signals to recipient NK cells because of different (mismatched) HLA class I molecules (HLA I). This mimics the "missing self" for NK cells, and we show that this is sufficient to provoke EC damage in vitro and in vivo. Importantly, we find that the mTORC1 pathway in NK cells is mandatory for this effect. Preclinical studies using experimental mouse models of transplantation confirm the efficiency of the mTORC1 inhibitor rapamycin to prevent the development of NK cell-mediated histological lesions.

## Results

**Antibodies are not the only trigger for graft MVI.** In kidney transplantation, MVI is defined as the presence of innate immune effectors in the lumen of peritubular capillaries (peritubular capillaritis, ptc) and/or glomeruli (glomerulitis, g)[20]. We retrospectively reviewed all kidney graft biopsies performed at our University Hospital between September 2004 and September 2012 ($n = 2024$ in 938 patients) and identified 129 renal recipients with typical graft MVI (g + ptc ≥ 2). The clinical characteristics of these patients are presented in Supplementary Table 1.

Interestingly, only 54% (70/129) of these patients had detectable circulating anti-donor HLA antibodies (i.e. "typical AMR"; Fig. 1a).

Previous studies have shown that bona fide AMRs can be triggered by non-HLA antibodies directed against either minor histocompatibility alloantigens or autoantigens[1,21–25]. Screening of patients' sera for anti-angiotensin II type 1-receptor (AT1R) and anti-MHC class I polypeptide-related sequence (MIC), two types of non-HLA antibodies the pathogenicity of which has been demonstrated[26,27], showed that neither the titre nor the proportion of positive patients were increased in the MVI +anti-HLA DSA− group (Supplementary Fig. 1A). Although it is impossible to test for all non-HLA specificities, previous studies have demonstrated that only antibodies able to bind to the surface of graft ECs can have a deleterious impact[13]. Flow cytometric cross-match with activated HLA-matched ECs[28] was therefore used to screen patients' sera for non-HLA anti-endothelial cell antibodies (Supplementary Fig. 1B, C). These experiments identified six patients (6/129, 4.6%) for whom non-HLA anti-endothelial cell antibodies could account for graft MVI (Supplementary Fig. 1D, E, Fig. 1a). Based on these results, we concluded that in almost half of the cases (53/129, 41.1%; group MVI+DSA−), graft MVI was not caused by host humoral response.

**Antibody-independent MVI affects graft survival.** The binding of large amounts of antibodies to graft endothelium triggers the classical complement pathway[29] responsible for acute tissue injuries, which dramatically curtail graft survival[14,15]. However, even in the absence of complement activation, DSA can still recruit innate immune effectors, and this MVI has a detrimental impact on graft survival[16].

In our cohort, we observed that of the 70 renal recipients with typical AMR (i.e. circulating anti-HLA DSA and MVI on graft biopsy), the 40 patients whose DSA were able to activate the complement cascade in vitro (group MVI+DSA+C3d+, Fig. 1a), had the highest score for C4d deposition in graft biopsy (Fig. 1b) and the worst graft survival (Fig. 1c).

Interestingly, the 53 patients with antibody-independent MVI (group MVI+DSA−) had the same graft survival as the 30 patients with AMR, owing to non-complement activating DSA (group MVI+DSA+C3d−). Graft survivals of these two groups were significantly better than that of MVI+DSA+C3d+ patients, but significantly worse than that of a matched control cohort without MVI or DSA (group MVI−DSA−) as shown in Fig. 1c.

We conclude that regardless of whether it is antibody-dependent or -independent, MVI has the same detrimental impact on graft survival.

**NK cells in antibody-dependent and antibody-independent MVI.** The nature and number of immune cells infiltrating the renal allograft of patients with available biopsy material from MVI+ DSA+C3d+ ($n = 17$), MVI+DSA+C3d− ($n = 14$), and MVI+ DSA− ($n = 32$) groups were compared using the computer-assisted analysis of graft inflammation (CAGI) method[30]. This approach allows a precise quantification of the innate and adaptive immune cell subset density in the microcirculation and the tubulointerstitial compartment of a renal allograft (Supplementary Fig. 2A, B). Quadratic discriminant analysis efficiently separated MVI+DSA+ C3d+ and MVI+DSA− patients, but a major overlap was observed between MVI+DSA+C3d− and MVI+DSA− (Fig. 1d), which suggests that a common pathophysiological process is taking place in the renal allografts of these two groups.

Antibodies that are unable to activate the complement cascade can still recruit FcY receptor-expressing innate immune effectors, which are responsible for antibody-dependent cell-mediated cytotoxicity (ADCC), thus leading to chronic/subclinical AMR[6]. Seminal experimental[17,31] and clinical[32] studies have demonstrated that among innate immune effectors, NK cells are crucial for the development of complement-independent AMR lesions. In line

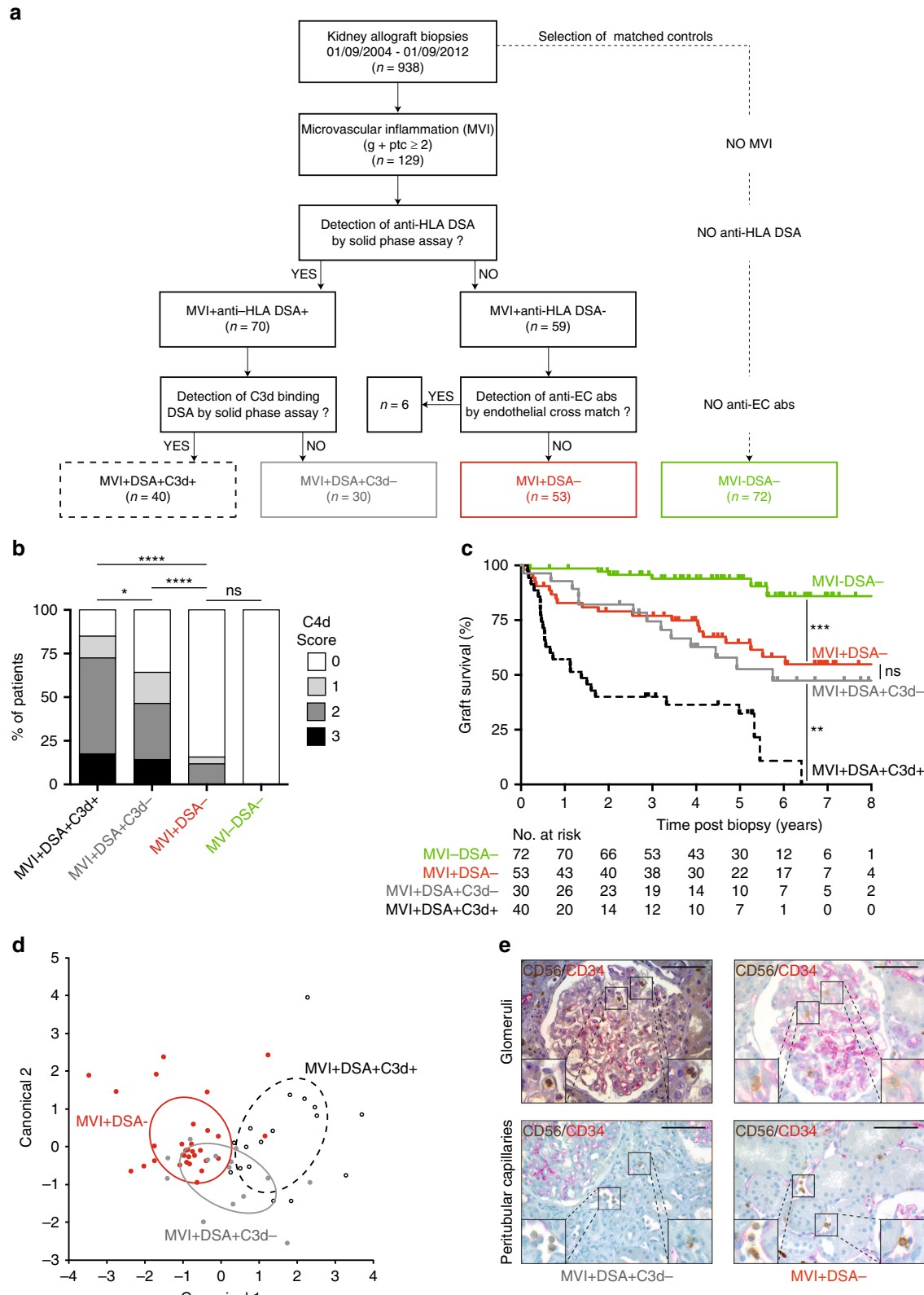

with this data, NK cells were present in the graft microcirculation of MVI+DSA+C3d− patients (Fig. 1e). Interestingly, NK cell infiltration was similar in MVI+DSA− patients, whose MVI was not triggered by antibody deposition on the graft endothelium (Supplementary Fig. 2B). We therefore concluded that in chronic vascular rejection, a final common pathway involving NK cells could be either triggered classically by the humoral arm of the

adaptive immune system of the recipients or induced by a direct antibody-independent activation of innate effectors.

**Missing self increases the risk of antibody-independent MVI.**
In steady state, the interaction of inhibitory KIRs with self-HLA I molecules of surrounding healthy cells provides a negative signal

**Fig. 1** Antibody-independent graft microvascular inflammation. Kidney-allograft biopsy material from 938 patients was screened for microvascular inflammation (MVI) lesions. Presence of circulating anti-HLA donor-specific antibodies (anti-HLA DSA) and their ability to activate the complement cascade was assessed in synchronous serum using solid-phase assays. Negative sera were additionally screened for anti-endothelial cell antibodies (anti-EC Abs) by flow cross-match (see Supplementary Fig. 1A). Three groups of patients were defined: group MVI+DSA+C3d+ ($n = 40$; dashed black line), group MVI+DSA+C3d− ($n = 30$; solid grey line), and MVI+DSA− ($n = 53$; solid red line). A fourth group (MVI−DSA−; $n = 72$; solid green line), devoid of MVI lesions in graft biopsy and circulating donor-specific antibody but matched for clinical characteristics with MVI+DSA− patients, was established. **a** Flow chart showing the distribution of the patients in the different groups. **b** C4d staining was quantified (0–3) in graft biopsy according to the Banff classification. Distribution of this parameter is shown for the four groups. $^{ns}p > 0.05$; $^*p < 0.05$, $^{****}p < 0.0001$; one-way ANOVA. **c** Renal graft survival curves were compared in the four groups. $^{ns}p \geq 0.05$, $^{**}p < 0.01$, $^{***}p < 0.001$; log-rank test. **d** Scatter plot of the first two canonical discriminant function analyses for CAGI dataset of kidney graft biopsies. Entropy $r^2 = 0.555$. **e** Paraffin-embedded sections of graft biopsies of patients from MVI+DSA+C3d− and MVI+DSA− groups were stained for endothelial cells (anti-CD34) and NK cells (anti-CD56). Representative images of MVI lesions found in the glomeruli (upper row) and the peritubular capillaries (lower row) are shown. Scale bar: 100 μm.

to NK cells (a graphical summary of this information is presented in Supplementary Fig. 3A)[33,34]. On the contrary, the down-regulated expression of HLA I molecules associated with tumoral transformation or viral infection triggers NK cell activation, which results in the destruction of the target cell, a process named response to "missing self". Of note, because the HLA locus is located on chromosome region 6p21, whereas the KIR locus is on 19q13.4, HLA and KIR are inherited independently. NK cells need to undergo a process of education, in which auto-reactive NK cells (owing to the lack of expression of HLA I ligands for their inhibitory KIRs) are rendered anergic (Fig. 2a).

We reasoned that although graft ECs express a normal level of HLA I molecules (Fig. 2b, Supplementary Fig. 4A), their allogeneic nature could theoretically induce a situation in which donor ECs express an HLA I allotype that is unable to interact with an educating inhibitory KIR expressed by the recipient's NK cells (i.e. "pseudo-missing self"; Fig. 2a). To test whether this hypothesis could explain antibody-independent graft MVI, we integrated, for each donor/recipient pair, the genetic analyses of recipient inhibitory KIRs and recipient HLA I (Supplementary Table 2) in order to identify educating inhibitory KIRs. Recipient data were then combined with the donor HLA I genotype (Supplementary Table 2) to identify situations of missing self. In line with our hypothesis, recipients with antibody-independent MVI had statistically more genetically predicted missing self (MS) than matched controls (Fig. 2c).

Of note, ~1/3 (15/43; 34.9%) of MVI+DSA− patients had no genetically predicted missing self, indicating that other molecular mechanisms can induce antibody-independent MVI (Fig. 2c). NK cell activation is governed by the integration of the signals provided by inhibitory and activating KIRs[35]. It is therefore tempting to speculate that in some patients, the activation of recipients' NK cells by graft endothelium was triggered by signalling through activating KIRs instead of (or in addition to) a lack of signalling through inhibitory KIRs. We were, however, unable to confirm this hypothesis, because the number and distribution of activating KIR genes were similar between MVI−DSA− and MVI+DSA− patients (Supplementary Table 2), and increasing the expression of KIR-activating ligands on ECs was neither necessary nor sufficient to trigger NK cell activation and promote EC damage (Supplementary Figs. 5A and 6A). These results, however, do not rule out the role of activating receptors in the triggering of NK cell-mediated rejection. Beyond activating KIRs, many other types of activating receptors exist on NK cells (including, NKG2D, NKG2C, NKp46, NKp3, etc.).

**Effect of priming and heterogeneity of the NK cell population**. Genetic analyses of the donor/recipient pairs also revealed that ~1/3 (21/55; 38.2%) of MVI−DSA− recipients had genetically predicted missing self, suggesting that this condition alone is not sufficient to trigger graft MVI (Fig. 2c).

In contrast to long-held belief, NK cells are not naturally active killers. Recent experimental evidence has demonstrated that educated NK cells need to undergo priming in order to acquire their full effector functions[36–38]. In clinical transplantation, two frequent situations can promote the priming of recipient's NK cells: ischaemia/reperfusion injuries and viral infections. Cold ischaemia time was longer in MVI+DSA− than in MVI−DSA− patients (Fig. 3a) and the incidence of viral infections, in particular cytomegalovirus infection, was higher in MVI+DSA− patients (Fig. 3b). These data suggest that the absence of graft MVI in some patients with genetically predicted missing self can be explained by the absence of sufficient priming of NK cells.

Another possible explanation for the absence of graft MVI in patients with genetically predicted missing self could be the inter-individual heterogeneity of the NK cell population. Because of the retrospective nature of our study and the lack of frozen peripheral blood mononuclear cells (PBMC), we were unable to test this hypothesis directly in the cohort. Instead, we performed a flow cytometry phenotypic analysis of the circulating NK cells of six healthy volunteers with identical inhibitory KIR genotypes (Supplementary Fig. 3B). The absolute count of NK cells among circulating lymphocytes showed huge inter-individual differences (Fig. 3c, Supplementary Fig. 4B). Beyond these quantitative disparities, variegated expression of KIR genes resulted in major inter-individual qualitative differences in the repertoire of inhibitory KIRs expressed by circulating NK cells (Fig. 3d). This had two main consequences: a high proportion (38.5%) of circulating NK cells did not express any inhibitory KIR (Supplementary Fig. 3C). Furthermore, even for inhibitory KIR-expressing NK cells, the proportion of NK cells able to specifically detect alteration of a particular HLA I molecule differed widely between the individuals (Fig. 3d). Therefore, it is conceivable that for some recipients with genetically predicted missing self and appropriate priming, the lack of MVI was due to the fact that the NK cell population that expressed the appropriate inhibitory KIR was too small.

The complexity of the clinical setting, in which each donor/recipient pair was different, at best allows only correlations. To confirm the causality of missing self in the occurrence of NK-mediated chronic vascular rejection of solid organ transplants we turned to experimental approaches.

**Allogeneic ECs trigger MS-induced NK cell activation in vitro**. To test whether endothelial missing self could trigger NK cell activation, primary allogeneic human ECs were co-cultured with NK cells purified from the PBMCs of healthy volunteers. After 4 h of culture, NK cells were recovered and their inhibitory KIR phenotype and activation status (i.e. expression of CD107a and MIP-1β) was assessed at the single-cell level by flow cytometry.

Our first analysis focused on the 5 NK cell populations that expressed only one inhibitory KIR (Supplementary Fig. 3B).

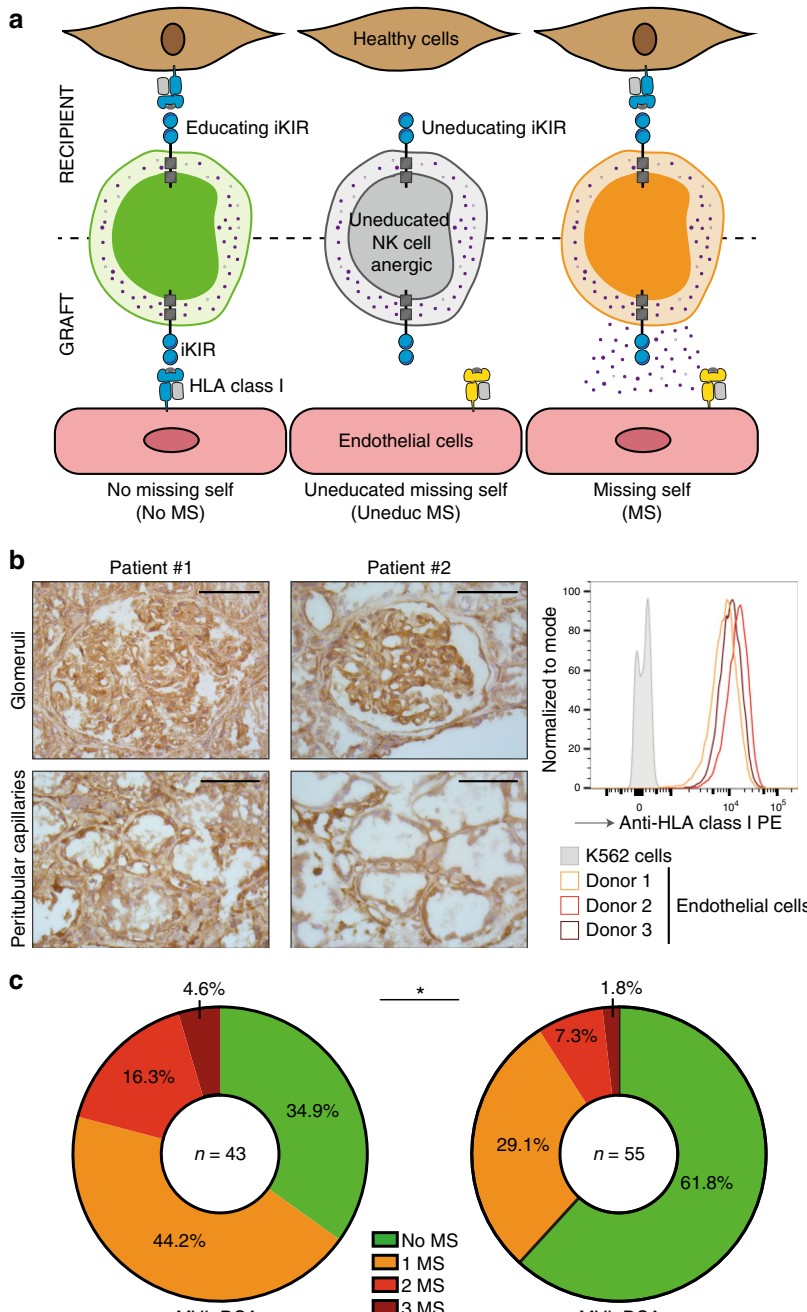

**Fig. 2** Genetically predicted missing self correlates with MVI. **a** Schematic representation of the education process of NK cells (upper row) and the three distinct situations that can be encountered by circulating NK cells when contacting the vasculature of an allogeneic organ (lower row): absence of missing self (no MS) (left panel), presence of a missing self for a ligand not expressed by the NK cell donor (uneducated missing self, uneduc MS) (centre panel), or missing self (MS) (right panel). **b** Left panel: representative images showing the expression of HLA I molecules on glomeruli (upper row) and peritubular capillaries (middle row) of renal graft biopsies of two distinct patients. Scale bar: 100 μm. Right panel: the expression of HLA I molecules was quantified on the surface of primary human endothelial cells from three different donors. K562 cells, which lack the expression of HLA molecules, were used as a negative control (grey). Representative flow cytometry profiles are shown. **c** Genetic analyses were conducted for each donor/recipient pair from the MVI +DSA− group (left panel) and MVI−DSA− (right panel, control), to identify situations of missing self (MS), in which allogeneic graft endothelial cells are unable to deliver the inhibitory signal to an educating inhibitory KIR of the donor. *$p < 0.05$; Fisher's exact test.

According to the HLA I genotypes of the ECs and NK cell donors, three distinct situations were identified for each of these NK cell populations (Fig. 2a): absence of missing self (no MS), presence of missing self for a ligand not expressed by the NK cell donor (uneducated missing self, uneduc MS), or missing self (MS). In line with the clinical data presented above, the three groups of NK cells behaved uniformly and showed no any sign of activation after co-culture with the ECs in the absence of prior priming (Supplementary Fig. 5A). After priming with low-dose IL2, NK cells that could specifically detect the absence of expression of a particular HLA I molecule (MS group) expressed significantly higher levels of both CD107a (Fig. 4a) and MIP-1β (Fig. 4b) compared to NK cells that did not express the specific inhibitory KIR (no MS) or that expressed the appropriate inhibitory KIR but

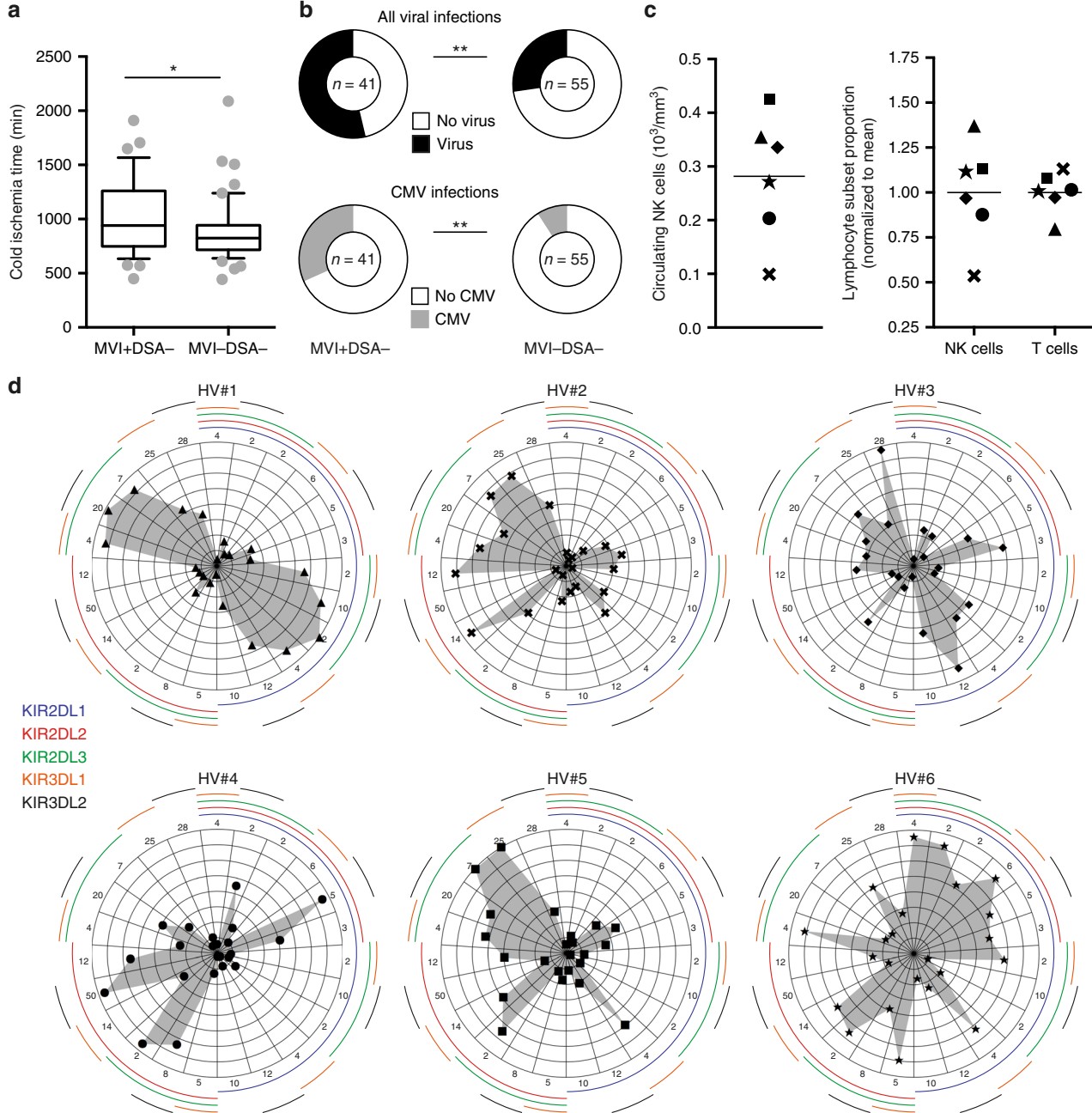

**Fig. 3** Influence of priming and heterogeneity of the NK cell population. **a** Cold ischaemia time was compared for patients from the MVI+DSA− and MVI−DSA group. The centre line in the boxes shows the medians; the box limits indicate the 25th and 75th percentiles, the whiskers indicate the 10th and 90th percentiles. *$p < 0.05$; Mann–Whitney test. **b** The prevalence of viral (all viruses, upper row) and cytomegalovirus (CMV, lower row) infections were compared for patients from MVI+DSA− (left column) and MVI−DSA− (right column) groups. *$p < 0.05$; **$p < 0.01$; Fisher's exact test. **c, d** Phenotypic analyses of circulating NK cells from six healthy volunteers (HV #1 to #6) with identical inhibitory KIR genotypes. **c** Left panel: individual values for absolute count of circulating NK cells are plotted. Right panel: the dispersion of the size of NK and T lymphocyte populations in the circulation of the 6 HV is shown. **d** Expression of the five inhibitory KIRs was assessed at the single cell level. The differences in distribution of the 23 NK cell subsets are shown for the six healthy volunteers. Each axis of the radar plot represents a given combination of inhibitory KIR as indicated by the colour code on the left. The scale of each axis is adjusted to optimise the display of every population of NK cells. Source data are provided as a Source Data file.

were not educated (uneduc MS). This result validates our hypothesis that allogeneic ECs can trigger missing self-induced activation of educated NK cells only if they are primed.

To determine whether some molecular combinations were more prone than others to promote missing self-induced NK cell activation, the previous dataset was re-analysed considering each inhibitory KIR separately. All inhibitory KIRs were equally able to promote activation of primed and educated NK cells in the

absence of their specific HLA I molecule on the ECs, except for KIR3DL2 (Supplementary Fig. 5B). This result fits the conclusion of recent independent reports[39,40] and suggests that KIR3DL2 might not be an educating inhibitory KIR. Taking this into account, we re-analysed the clinical data presented in Fig. 2c, removing KIR3DL2 from the definition of genetically predicted missing self. Although this simple change did not correct the other limitations of this method (details discussed in the section

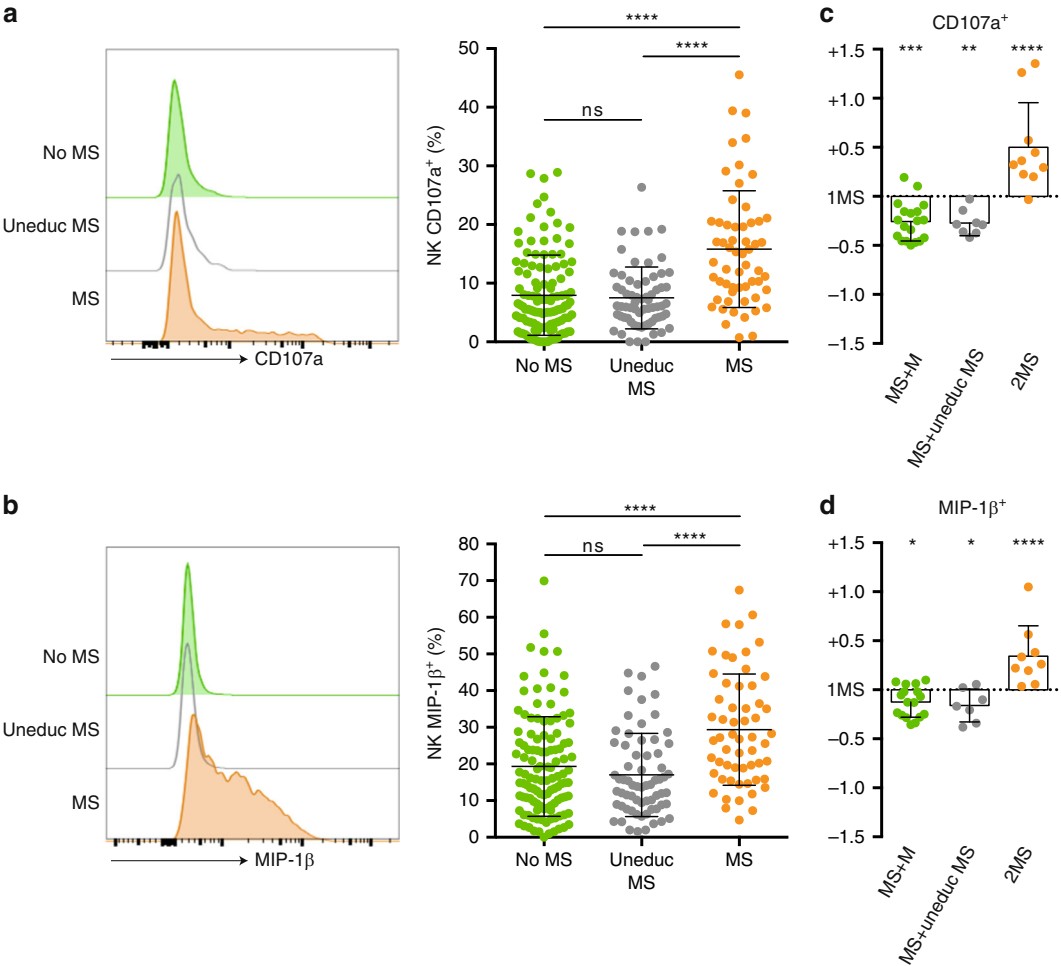

**Fig. 4** Allogeneic endothelial cells trigger missing self-induced activation of NK cells. Primary allogeneic human endothelial cells were co-cultured with purified NK cells from 30 healthy volunteers primed with low-dose IL-2. After 4 h of culture the activation status of the NK cells was assessed at the single cell level by flow cytometry. **a**, **b** Analyses were focused on the five NK cell populations that expressed a single inhibitory KIR. **a** Expression of CD107a (LAMP-1) on NK cell surface after 4-h co-culture. Left panel: representative flow cytometry profiles. Middle panel: individual values of NK cell populations according to their status against primary allogeneic human endothelial cells. **b** Intracellular staining for MIP-1β in NK cells after 4-h co-culture. Left panel: representative flow cytometry profiles. Middle panel: individual values of NK cell populations according to their status against primary allogeneic human endothelial cells. **c**, **d** Analyses were focused on the NK cell populations that expressed two inhibitory KIRs, one of them lacking its ligand on endothelial cells (missing self, MS). According to the nature of the second inhibitory KIR, three situations were distinguished: 1MS+1M, if endothelial cells expressed the ligand for the second inhibitory KIR; 1MS + uneduc MS, if neither endothelial cells nor NK cell donor expressed the ligand for the second inhibitory KIR, and 2MS, if endothelial cells did not express the ligands of the two inhibitory KIRs. Results are normalised over the value observed for the NK cell population that expressed the single mismatched inhibitory KIR. **c** Expression of CD107a (LAMP-1) on NK cell surface after 4-h co-culture. **d** Intracellular staining for MIP-1β in NK cells after 4 h co-culture. ns:$p \geq 0.05$, *$p < 0.05$, **$p < 0.01$, ***$p < 0.001$, ****$p < 0.0001$; one-way ANOVA. Source data are provided as a Source Data file.

"Impact of priming and heterogeneity of NK cell population"), it was sufficient to reduce the proportion of recipients with genetically predicted missing self in the group without graft MVI [proportion of recipients with genetically predicted missing self in MVI-DSA- group with vs. without KIR3DL2: 21/55 (38.2%) vs. 15/55 (27.3%)].

A significant proportion of NK cells (25.4%, Supplementary Fig. 3C) express more than one inhibitory KIR on their surface. To determine how these distinct signals contribute to cell activation, we focused the analysis on NK cells that expressed two inhibitory KIRs, one being responsible for missing self-induced activation. Depending on the second inhibitory KIR of the NK cell and the HLA I genotypes of the ECs, three situations were identified: missing self + matched (MS + M), missing self + uneducated missing self (MS + uneduc MS) or missing self + missing self (2MS). Activation status of the NK cells of these three

groups after co-culture with allogeneic ECs was compared to that of NK cells expressing only one educating inhibitory KIR. The level of expression of both CD107a and MIP-1β was increased in 2MS, and decreased in MS + uneduc MS and the MS + M group (Fig. 3c, d). These results demonstrate that the signals generated by each educating inhibitory KIR expressed on the surface are integrated by NK cells and modulate missing self-induced activation.

**MS-induced NK cell activation damage allogeneic ECs.** Having demonstrated that allogeneic ECs could trigger missing self-induced NK cell activation, we sought to determine its impact on ECs. To this end, the integrity of adherent ECs was monitored by real-time impedance measurement in the co-culture model described above.

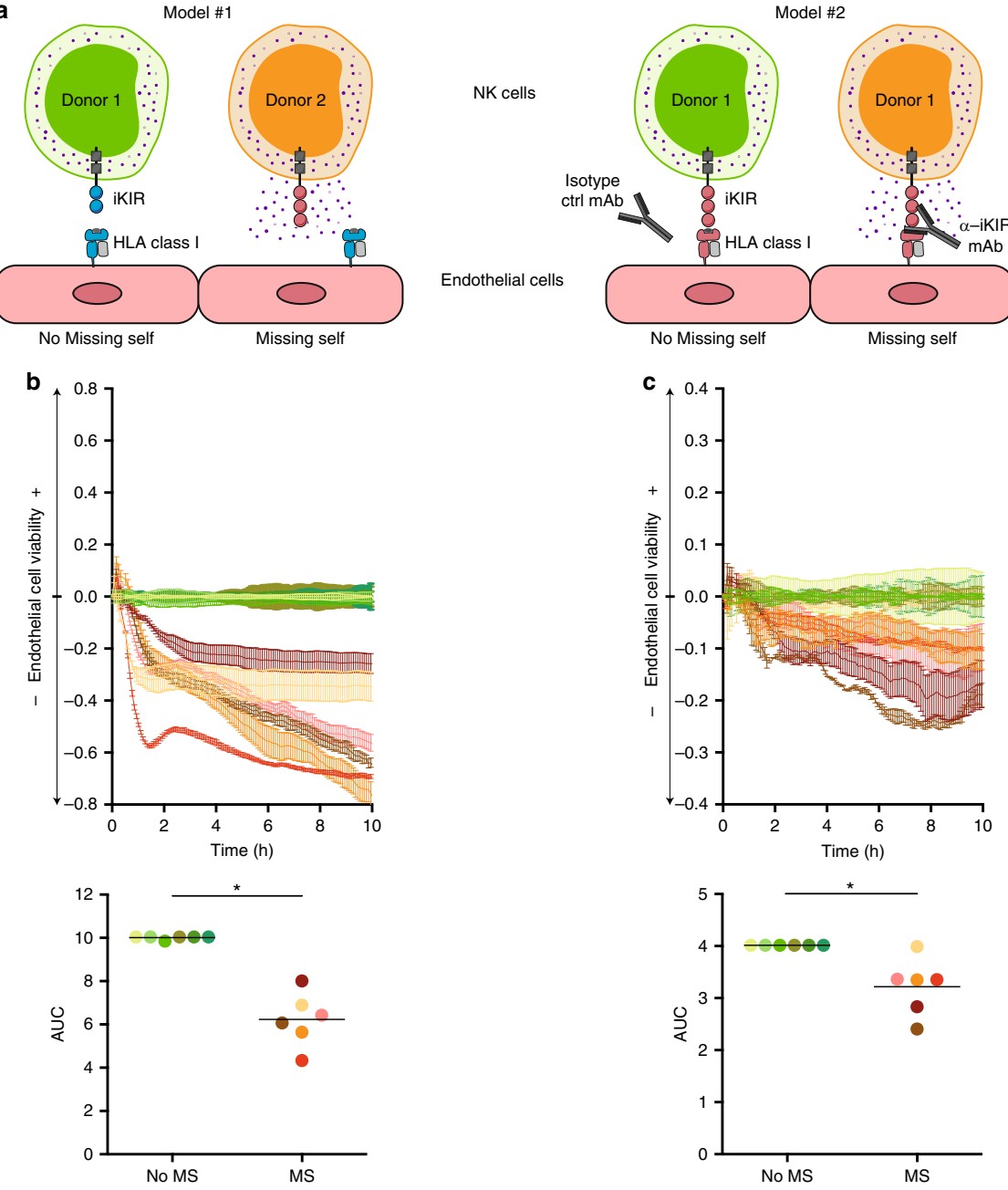

**Fig. 5** Missing self-induced NK cell activation is harmful for endothelial cells. **a** Schematic representation of the two experimental models. Purified NK cells from healthy volunteers were co-cultured with adherent primary allogeneic human endothelial cells, the viability of which was assessed by real-time impedance measurement. Real-time impedance data of the experimental co-culture were normalised over control. **b** Model #1: The same primary human endothelial cells were co-cultured with NK cells purified from two distinct donors: the first without missing self (control co-culture) and the second with missing self (experimental co-culture). Individual impedance profiles (mean ± standard error; upper panel) and the area under the curves (lower panel) from six independent experiments are shown. **c** Model #2: The same primary human endothelial cells were co-cultured with NK cells purified from a donor without missing self, in the presence of a blocking anti-inhibitory KIR mAb (experimental co-culture) or an isotype control mAb (control co-culture). Individual impedance profiles (mean ± standard error; upper panel) and area under the curves (lower panel) from six independent experiments are shown. *p < 0.05; Wilcoxon signed rank test.

In a first set of experiments (model #1, presented in Fig. 5a), we compared the survival of the same primary allogeneic ECs exposed to NK cells from two distinct donors: one with missing self and the other without (negative control). The experiment, reproduced with six different pairs, demonstrated that endothelial cell survival was consistently reduced when co-cultured with NK cells expressing one inhibitory KIR unable to interact with the appropriate HLA I molecules on ECs (Fig. 5b).

To rule out the possibility that the differences observed in the first model were influenced by inter-individual heterogeneity of NK cell populations between donors, we developed a second model (model #2, presented in Fig. 5a), in which the allogeneic ECs were co-cultured with NK cells from the same matched donor with anti-KIR3DL1 blocking mAb or an isotype control mAb. In line with previous results, co-cultures with anti-KIR3DL1 blocking mAb induced an "artificial" missing self,

which significantly lowered the endothelial cell survival rate (Fig. 5c).

Collectively, these in vitro data support the notion that missing self-induced NK cell activation has a deleterious impact on graft vasculature.

**MS triggers NK cell-mediated rejection in vivo.** We next investigated the impact of missing self-induced NK cell activation in vivo in the context of transplantation. We adapted the heterotopic heart transplantation model as shown in Fig. 6a. Heart grafts, harvested in wild-type C57B/L6 (controls) or ß2-microglobulin KO mice, were transplanted to wild-type C57B/L6 mice. As observed in the clinic, the mere absence of MHC I molecules on the graft endothelium was insufficient to promote the development of histological lesions (group ß2-microglobulin KO into C57B/L6, ß2→B6; Fig. 6b, d). However, the priming of recipients' NK cells induced by mild ischaemia/reperfusion injuries resulted in the appearance of MVI, specifically in ß2-microglobulin KO heart transplants (groups ß2-microglobulin KO into C57B/L6 + ischaemia, ß2→B6 + isch vs. C57B/L6 into C57B/L6 + ischaemia, B6→B6 + isch; Fig. 6b, d). Similar results were obtained when the priming of NK cells was performed with Poly (I:C), a surrogate for viral infection (Supplementary Fig. 7B, C). Graft MVI in this model was similar to that observed in MVI+DSA− patients: circulating CD45+ immune cells, including Nkp46+ NK cells, were found to adhere to CD31+ turgid capillary ECs (Fig. 6c, d). The central role of NK cells in this type of rejection was demonstrated by the complete disappearance of lesions in ß2-microglobulin KO heart grafts transplanted to recipients, whose NK cells were depleted by anti-NK1.1 mAb (group ß2-microglobulin KO into C57B/L6+ ischaemia + anti-NK1.1, ß2→B6 + isch + αNK1.1; Supplementary Fig. 7A, Fig. 6b–d).

**MS triggers mTORC1 signalling in NK cells.** To gain insight into the molecular mechanisms involved in missing self-induced NK cell activation, purified human NK cells of healthy volunteers were co-cultured with K562 cells, an MHC I-deficient human cell line.

Based on previous work from our group[41,42], the analysis focused on the mTOR pathway. The phosphorylation status of S6 ribosomal protein (S6RP) and protein kinase B (Akt), located downstream from mTORC1 and mTORC2 complexes, respectively, was longitudinally assessed in NK cells using imaging flow cytometry (Supplementary Fig. 8A, Fig. 7a). While isolated NK cells showed only a modest increase in p-S6RP, the mTORC1 pathway was strongly activated in NK cells that had formed doublets with K562 targets (Fig. 7a, b). By contrast, no significant change was observed regarding the phosphorylation status of Akt in NK cells, which suggests that mTORC2 does not play a role in missing self-induced NK cell activation (Fig. 7a, c).

Analysis of graft biopsies from patients with missing self-induced NK cell-mediated rejection confirmed that the mTORC1 pathway was activated in NK cells adherent to graft microvasculature (Fig. 7d).

**mTOR inhibitor prevents MS-induced NK-mediated rejection.** Based on the molecular data presented above and data from the literature[43–45], we hypothesised that mTOR inhibitors might have potent therapeutic effects against missing self-induced NK-mediated rejection.

First, the ability of the mTOR inhibitor rapamycin to block the mTORC1 pathway and suppress missing self-induced cytotoxicity of human NK cells was evaluated ex vivo. NK cells were purified from the circulation of 24 patients before and 1 month after

introduction of the mTOR inhibitor and the level of phosphorylation of S6 ribosomal protein (S6RP, which is located downstream mTORC1) was measured by flow cytometry. Exposure to the drug in vivo not only decreased the baseline level of phosphorylation of S6RP in NK cells, but also drastically reduced their response to the stimulation by IL-15 (Fig. 8a, Supplementary Fig. 4C). As expected from the above, adjunction of mTOR inhibitor to co-cultures of K562 cells and human NK cells reduced missing self-induced cytotoxicity (Fig. 8b).

To further validate the therapeutic potential of mTOR inhibitor in missing self-induced NK-mediated rejection, the effects of an mTOR inhibitor and a calcineurin inhibitor were compared in the in vivo mouse model (Fig. 8c). In line with our hypothesis, calcineurin-inhibitor-treated animals developed the same MVI as untreated controls, but recipient mice treated with an mTOR inhibitor showed significantly less endothelial turgidity and inflammatory effectors in heart graft capillaries (Fig. 8d–f).

Our data therefore validate the idea that therapeutic mTOR inhibition may protect transplant recipients against missing self-induced NK-mediated rejection.

## Discussion

In the present study, we demonstrate that the allogeneic nature of graft ECs sometimes creates "missing self", a situation that can be sensed by primed NK cells in the recipient's circulation. Missing self-induced NK cell activation promotes the development of graft MVI that has the exact same detrimental impact on organ survival as non-complement activating anti-HLA DSA, the primary cause of late transplant loss[6–8]. However, while there is currently no efficient therapy against antibody-mediated chronic vascular rejection, our study established that missing self-induced NK cell activation is dependent upon the mTORC1 pathway and can be blocked by mTOR inhibitors, a commercially available class of immunosuppressive drugs[46]. Preclinical studies, using experimental murine models, suggested that therapeutic mTORC1 inhibition can prevent the development of histological lesions.

We believe that this data can have several levels of significance. Firstly, clinicians in charge of transplant patients are frequently confronted with MVI lesions on graft biopsy. As an illustration, the prevalence of MVI lesions was estimated to be as high as 13.8% in our cohort of renal transplant patients, although we do not perform HLA incompatible transplantations (i.e. transplantation in the presence of preformed anti-HLA DSA) in our centre. Until now, MVI lesions have been considered as the hallmark of AMR. Our data instead suggest that in half of the cases, MVI is the result of an unrecognised type of rejection due to the "direct" activation of recipient's NK cells by missing self.

An intriguing question is why previous large clinical studies have failed to detect the detrimental impact of genetically predicted KIR-ligand incompatibility on renal allograft survival[47,48]. Our results clearly show that genetically predicted missing self is not a sufficient condition to develop NK cell-mediated rejection. Firstly, patients with similar genetic profiles exhibit high inter-individual heterogeneity in the size of the NK cell population able to sense the missing self. Secondly, even among patients (and mice) with sufficient NK cells able to sense the missing self, only those whose NK cells were previously primed (by ischaemia/reperfusion injuries or a viral infection) went on to develop MVI lesions. Finally, another layer of complexity may arise from the fact that NK cells have also been shown to regulate the alloimmune response through the killing of donor antigen-presenting cells[49–51].

Although missing self-induced and non-complement-activating anti-HLA AMRs have the same detrimental impact on graft

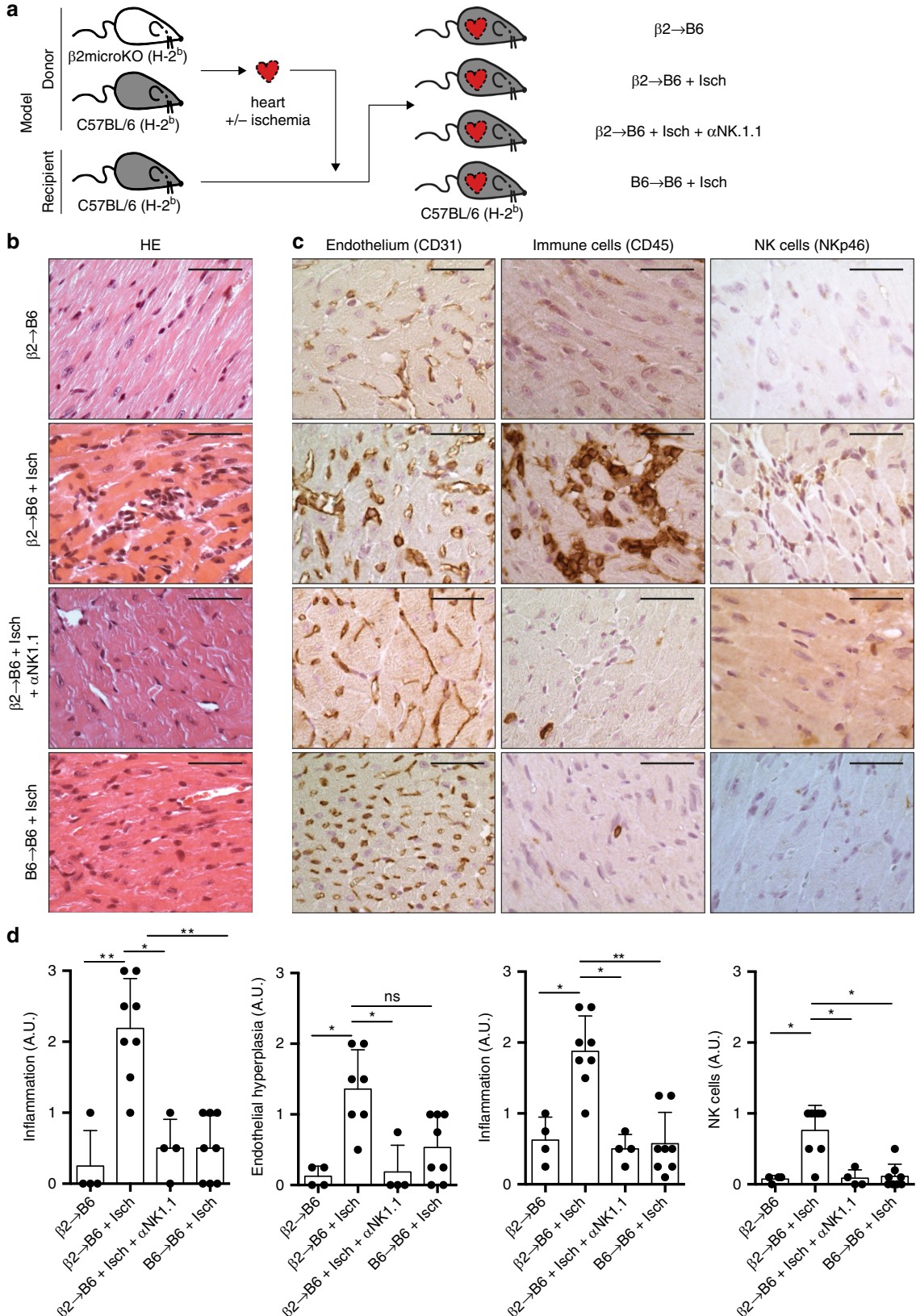

survival, it is crucially important to differentiate these two conditions. Patients with missing self-induced rejection will not respond to the costly and tedious treatment of AMR, which associates plasmapheresis with high-dose intravenous immunoglobulins[6]. The mixture of authentic cases of AMR with

previously unrecognised cases of missing self-induced rejection might explain the high heterogeneity in response to treatment[6,15]. Furthermore, our results demonstrate that therapeutic mTORC1 inhibition efficiently prevents the development of histological lesions due to missing self-induced NK cell activation in a murine

**Fig. 6** Missing self triggers NK cell-mediated rejection in vivo. **a** Schematic representation of the murine experimental models. Wild-type C57BL/6 (B6) mice were transplanted with either a C57BL/6 or a β2-microglobulin KO heart (β2). In some cases, the heart was subjected to 3 h of cold ischaemia before transplantation (+Isch). Some recipients were treated with anti-NK depleting mAb (+αNK1.1). Heart grafts were harvested 60 days after transplantation for histological analysis. **b** Representative findings of H&E stain are shown for the four experimental groups (two independent experiments): from top to bottom β2→B6 (n = 4); β2→B6 + Isch (n = 8); β2→B6 + Isch + αNK1.1 (n = 5); and B6→B6 + Isch (n = 8). Scale bars: 100 μm. **c** Immunohistochemistry was performed to evaluate the morphology of the microvasculature (CD31), the immune cell infiltration (CD45), and the NK cell infiltration (Nkp46). Representative findings are shown for the four experimental groups (two independent experiments). Scale bars: 100 μm. **d** A trained pathologist blindly graded the intensity of each elementary lesion on a semi-quantitative scale (score 0–3). Mean ± standard deviation. *p < 0.05; **p < 0.01; one-way ANOVA.

experimental model of transplantation. Of note, on the basis of conflicting clinical reports suggesting that mTOR inhibitors might be less potent in preventing DSA generation[52–55]. transplanted patients with graft MVI (and wrongly diagnosed with AMR) are often switched from an mTOR inhibitor-based to calcineurin-inhibitor (CNI)-based maintenance regimen. This is probably detrimental to graft survival because our data show that CNI have no impact on missing self-induced rejections. A note of caution should, however, be sounded in the case of renal transplantation. NK cell-induced microvascular lesions in the glomeruli may trigger significant stress on podocytes. These key cellular players of the filtration barrier are known to adapt to stress through mTOR pathways[56,57] and it is therefore possible that introduction of mTOR inhibitors late in the course of the disease (i.e. in the presence of proteinuria and/or chronic glomerular lesions on the biopsy) could be ill-tolerated in these patients.

Beyond its clinical impact, our work is also of interest for basic immunologists. Until very recently, rejection of allografts was thought to be strictly dependent on the recipient's adaptive immune system. The consensus sequence described in textbooks starts with the recognition of donor-specific HLA molecules by the recipient's T lymphocytes through direct or indirect pathways[58,59]. Direct allorecognition of donor-specific HLA molecules as intact complexes on the surface of passenger leucocytes activates up to 10% of a recipient's T cells, which triggers acute cellular rejection. By contrast, the "indirect" recognition of allogeneic peptides, presented within MHC-II molecules on the surface of the recipient's APCs, activates much fewer CD4+ T cells, but these are critically important for the generation of alloantibodies[60]. In this prevalent model, innate immune cells in general (and NK cells in particular) are merely considered as downstream effectors that participate in the destruction of the graft only upon recruitment by the adaptive immune system[31,61,62].

Our present work challenges this vision and coincides with the concept of innate allorecognition[63], which proposes that the innate immune system alone can promote rejection of transplanted organs. The first experimental evidence supporting this notion came in the 1960s from the observation that bone marrow grafted from parental strains of mice to F1 hybrids between the parental strain and a second strain was rejected, a process referred to as "hybrid resistance"[64]. Hybrid resistance was later linked to the ability of NK cells to react to missing self[65], but for years this process was not thought to be involved in solid tissue rejection because Snell's third "law of transplantation" stated that "skin grafts from either inbred parent strain to the F₁ hybrid succeed"[66], and MHC homozygous embryonic stem cell-derived teratomas form and persist in MHC heterozygous mice[67]. We believe that the lack of impact of missing self-induced NK cell activation in these two models is due to the fact that in both cases the graft vasculature comes from the recipient[13] and therefore has a "normal" expression of self-MHC. Our hypothesis is consistent with the seminal work by Uehara et al.[68] (who reported that missing self-induced NK cell activation leads to the development of chronic allograft vasculopathy with no accompanying interstitial inflammation in parental cardiac grafts transplanted to F1

hybrid recipients), and the fact that histological lesions were limited to the graft vasculature in both the patient biopsies and the murine experimental model in our own study.

Importantly, NK cells might not be the only innate immune effectors capable of innate allorecognition, as ~1/3 of MVI+ DSA− patients in our cohort had no genetically predicted missing self, suggesting that other innate immune effectors are able to induce antibody-independent graft MVI. Recent data from Fadi Lakkis's group identifies the recipient's monocytes as probable culprits. Monocytes are indeed able to distinguish between self and allogeneic non-self through the expression of CD47, a surface receptor able to sense SIRPα polymorphism in the donor[69]. In accordance with this theory, it has been reported that depleting macrophages from recipients of parental to F1 cardiac transplants did reduce the formation of chronic allograft vasculopathy in an murine experimental model[70].

In conclusion, this study identifies a type of chronic rejection, whose pathophysiology is independent of the recipient's adaptive immune system. Missing self-induced NK cell-mediated chronic vascular rejection is as prevalent as AMR and has the same detrimental impact on organ survival. However, while there is currently no efficient therapy against chronic AMR, commercially available mTOR inhibitors have shown promising efficacy in preventing the development of histological lesions in a preclinical murine model of missing self-induced NK cell-mediated chronic vascular rejection.

## Methods

**Human studies.** The study was carried out in accordance with French legislation on biomedical research and the Declaration of Helsinki. All patients gave informed consent for the utilisation of clinical data [Données Informatiques Validées en Transplantation (DIVAT)] and biological samples for research purpose. For DIVAT, a declaration was made to the CCTIRS (Comité consultatif sur le Traitement de l'Information en matière de Recherche dans le domaine de la Santé) and the CNIL (Commission nationale de l'Informatique et des Libertés). For the biocollection, an authorisation (No. of biocollection: AC- 2011-1375 and #AC-2016-2706) was obtained from the French Ministry of Higher Education and Research (direction générale pour la recherche et l'innovation, cellule bioéthique).

The computer database (DIAMIC) of the Lyon University Hospital pathology department was used to screen all kidney-allograft biopsies (2024 biopsies in 938 patients) performed between 1 September 2004 and 1 September 2012 for microvascular inflammation (MVI+). The biopsies of the 143 patients were systematically reviewed by the same trained pathologist (M. Rabeyrin), who graded the lesions according to the Banff 2011 classification. Fourteen patients, whose biopsy analysis did not confirm the presence of MVI lesions (Banff g + ptc score < 2) were excluded. Computer-assisted analyses were conducted to quantify T cells, B cells, granulocytes, macrophages and NK cells in the patient biopsies.

Clinical data of the 129 patients enrolled in the study obtained with DIVAT were crosschecked with the CRISTAL database [Cristal: http://www.sipg.sante.fr/portail/]. The patient characteristics are summarised in Supplementary Table 1.

Serum samples banked at the time of biopsy were screened for the presence of anti-HLA DSA, and, if positive, for the ability of these anti-HLA DSA to bind the complement fraction C3d. These centralised analyses were performed in a blinded fashion with single-antigen flow bead assays according to the manufacturer's instructions (Immucor, Norcross, GA, USA). If negative, all the serum samples collected during the follow-up of the patients were checked to confirm this negativity. To rule out the presence of non-HLA donor-specific antibodies, negative sera were tested in endothelial flow cross-match assay.

The steps leading to the distribution of patients into the first three groups of patients (MVI+DSA+C3d+, n = 40; MVI+DSA+C3d−, n = 30; and MVI+ DSA−, n = 53) are summarised in Fig. 1a.

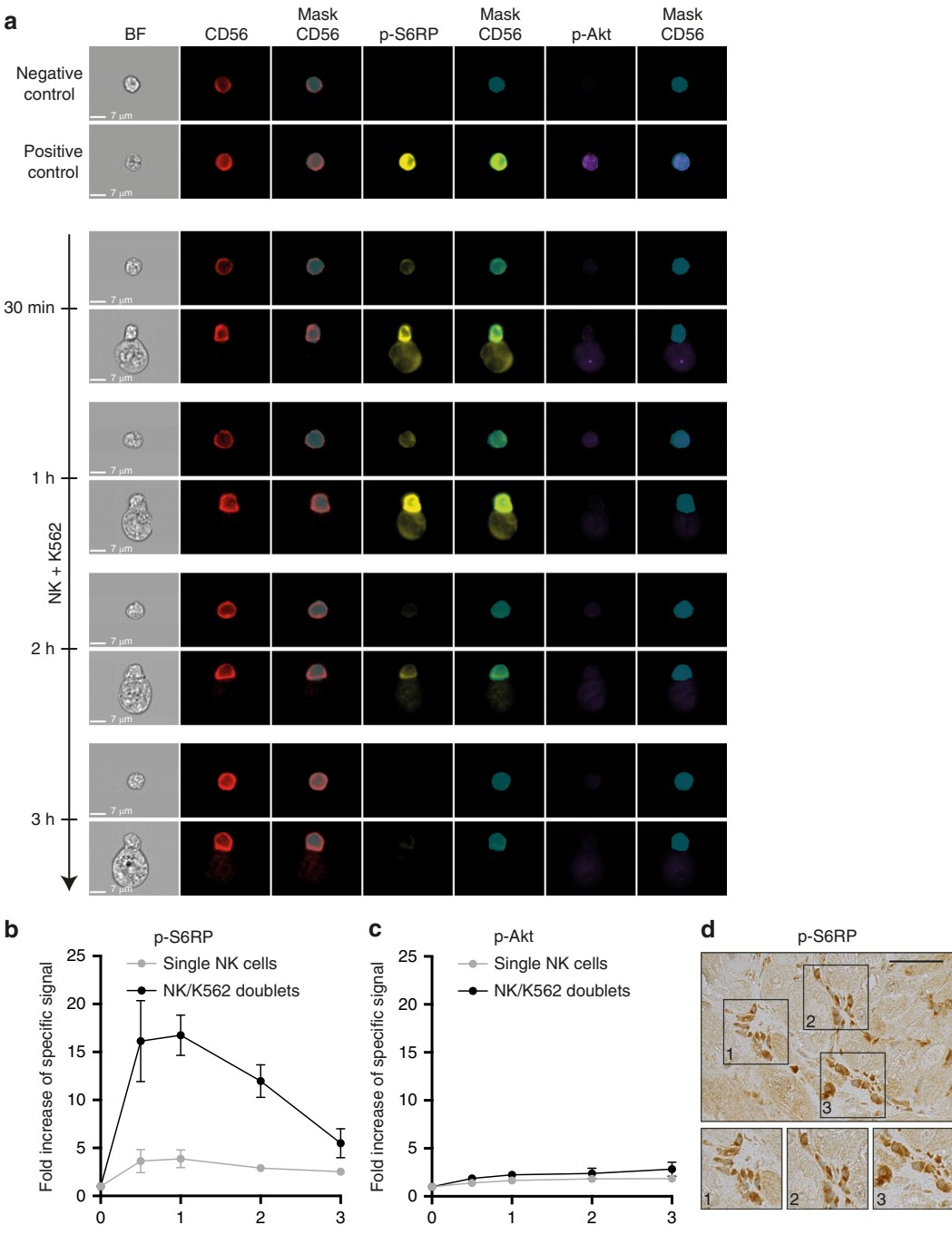

**Fig. 7** Missing self-induced NK cell activation is mTORC1-dependent. **a–c** Purified NK cells from a healthy donor were co-cultured with HLA-deficient K562 cells. An imaging flow cytometer was used at indicated time points to detect the phosphorylated form of S6 ribosomal protein (S6RP, downstream mTORC1) and of protein kinase B (Akt, downstream mTORC2), in isolated NK cells and NK cells that form doublets with K562 target cells. **a** Representative images of NK cells cultured alone (negative control), in the presence of IL-15 (positive control), and from co-cultures with K562 cells are shown. **b** The intensity of the signal corresponding to the phosphorylated form of S6RP was measured at various time points in NK cells (CD56 mask), isolated (grey curve) or in doublet with K562 cells (black curve). Data were normalised over baseline; mean ± standard deviation. **c** The intensity of the signal corresponding to the phosphorylated form of Akt was measured at various time points in NK cells (CD56 mask), isolated (grey curve) or in doublet with K562 cells (black curve). Data were normalised over baseline; mean ± standard deviation. **d** Graft biopsies of heart transplant patients diagnosed with missing self-induced NK-mediated rejection were stained for the phosphorylated form of S6 ribosomal protein (p-S6RP), which is located downstream mTORC1. A representative image is shown. Scale bars: 100 μm. Source data are provided as a Source Data file.

A control group, without MVI on graft biopsy, nor circulating DSA (MVI− DSA−, $n = 75$), but matched for the main clinical characteristics of the MVI+ DSA− patients, was established from the pool of 938 patients.

**Clinical pathology**. Kidney graft biopsies were performed systematically as part of the routine follow-up procedure at 3 months and 1-year post-transplantation, or if rejection was suspected at the other time points.

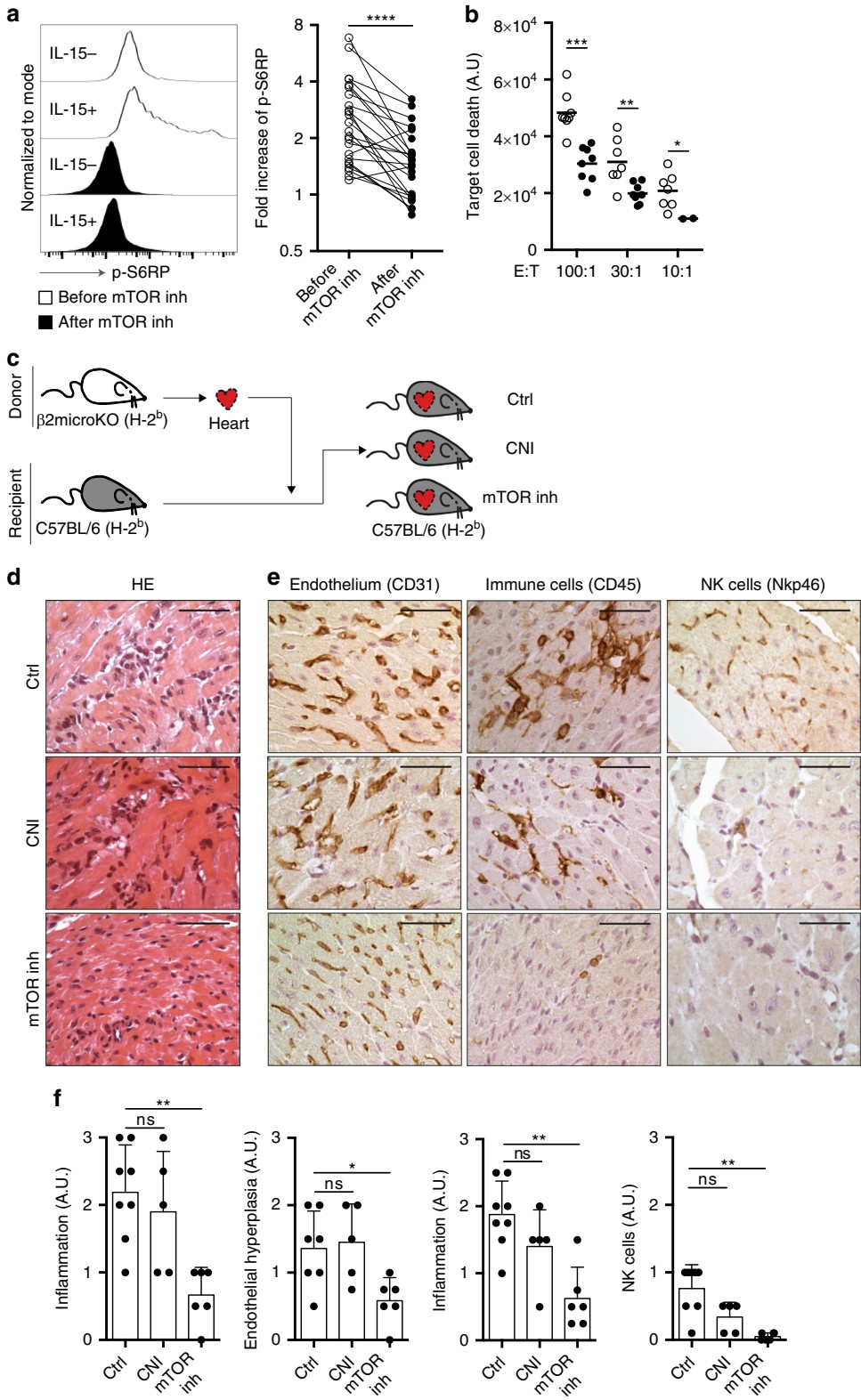

Renal specimens were fixed in acetic acid–formol–absolute alcohol, and paraffin-embedded sections were stained by routine methods. C4d staining was performed by indirect immunofluorescence on frozen sections using an anti-human C4d complement-rabbit clonal antibody (clone A24-T, produced by DB Biotech, Kosice, Slovak Republic).

The renal pathologist (M. Rabeyrin) who reviewed the biopsy specimens was blinded to clinical and immunological data.

For CAGI, double stainings with anti-CD34 (endothelial cells) and respectively one antibody among anti-CD3 (T cells), anti-CD20 (B cells), anti-CD66b (granulocytes), anti-CD68 (macrophages) and anti-CD56 (NK cells) were performed by immunochemistry on paraffin-embedded sections using an anti-human CD34 (clone QBEnd10, 1/200, Dako, Les Ulis, France) and respectively anti-human CD3 (clone SK7, 1/150, Becton Dickinson, Le Pont de Claix, France), anti-human CD20 (Clone L26, 1/400, Dako), anti CD66b (clone G10F5, 1/300,

**Fig. 8** mTOR inhibition prevents missing self-induced NK cell-mediated rejection in vivo. **a** PBMCs from 24 patients were collected before and 1 month after injection of an mTOR inhibitor and p-S6RP was quantified in NK cells stimulated (IL-15+) or not with IL-15 (IL-15−). Representative flow cytometry profiles are shown (left panel). The relative increase of p-S6RP signal in NK cells after stimulation with IL-15 is shown (right panel). Each circle is a patient. Results obtained before (open circles) and after mTOR inhibitor introduction (black circles) were compared. ****$p < 0.0001$; Wilcoxon signed rank test. **b** Purified NK cells from eight healthy volunteers were co-cultured with K562 cells at different effector-to-target ratios (E:T) for 6 h in presence (mTORinh+) or not (mTORinh−) of an mTOR inhibitor. Viability of target cells was compared. *$p < 0.05$, **$p < 0.01$, ***$p < 0.001$; Mann–Whitney test. **c–f** Wild-type C57BL/6 mice were transplanted with β2-microglobulin KO hearts subjected to 3 h of cold ischaemia. Recipient mice were treated with vehicle (control, Ctrl): a calcineurin inhibitor (CNI) or an mTOR inhibitor (mTORinh). Heart grafts were harvested 60 days after transplantation for histological analysis. Two independent experiments. **c** Graphical summary of the model. **d** Representative findings of H&E stain are shown for the three experimental groups: from top to bottom Ctrl ($n = 8$); CNI ($n = 5$); mTORinh ($n = 7$). Scale bars: 100 μm. **e** Immunohistochemistry was performed to evaluate the morphology of the microvasculature (CD31), the immune cell infiltration (CD45), and the NK cell infiltration (Nkp46). Representative findings are shown for the three experimental groups. Scale bars: 100 μm. **f** A trained pathologist blindly graded the intensity of each elementary lesion on a semi-quantitative scale (score 0–3). Mean ± standard deviation. ns:$p \geq 0.05$; *$p < 0.05$; **$p < 0.01$; ****$p < 0.0001$; one-way ANOVA. Source data are provided as a Source Data file.

Becton Dickinson), anti-human CD68 (clone PGM1, 1/100, Dako) and anti-human CD56 (clone CD564, 1/10, produced by Novocastra and distributed by Leica Microsystemes SAS, Nanterre, France). Of note, CD56 staining, although extremely sensitive (94%) is not fully specific for NK cells. About 3/4 of CD56+ cells are NK (Nkp46+). Most CD56+Nkp46− cells are either NKT or CD8+ T cells.

Computerised quantitative analyses were conducted to quantify the density of each immune cell type in the microcirculation and tubulointerstitial compartment of the renal allograft.

For heart graft biopsies, phosphorylated-S6RP staining was performed by immunohistochemistry on paraffin-embedded sections using an anti-human p-S6RP(Ser240/244) antibody (clone D68F8, produced by Cell Signaling Technology, Leiden, The Netherlands) according to Tible et al.[71].

**Detection of anti-HLA antibodies**. Serum samples, banked at the time of biopsy from patients with significant MVI, were diluted fivefold in washing buffer and tested for the presence of donor-specific anti-HLA antibodies using Screening Flow Beads (LifeScreen, Class I and Class II ID ®, Lifecodes, Immucor) and Single Antigen Flow Beads (LSA class I and class II®, Lifecodes, Immucor) in the case of positivity or a questionable result of the screening test. To rule out a false-negative result due to interference with complement proteins, all sera positive in Screening Flow Beads but without identified DSA in Single Antigen Flow Beads were retested after a pre-treatment with EDTA.

All the sera of MVI+DSA− and MVI−DSA− patients collected before the biopsy were also checked, and patients with circulating donor-specific anti-HLA antibodies detected at any time point were excluded.

All the analyses were performed in a blinded fashion by the same trained immunobiologist (V.D.) at the Etablissement Français du Sang, Lyon, France.

**Detection of non-HLA antibodies**. Sera of patients were screened for the presence of anti-AT1R and anti-MIC antibodies using a multiplex solid-phase assay (Immucor).

To detect non-HLA anti-endothelial cell antibodies, we used a flow cross-match technique. Briefly, target ECs were HLA-matched to avoid false-positive tests due to HLA binding (for sera containing anti-HLA antibodies that were not specific to kidney donor). Confluent endothelial cell monolayers were starved overnight in endothelial cell basal medium supplemented with 2% fetal bovine serum (FBS) without growth factors and incubated with recombinant human tumour necrosis factor α (TNF-α) (100 U/mL; Peprotech) for 48 h. ECs were then dissociated with trypsin, and $1–2 \times 10^5$ ECs were incubated for 30 min at room temperature with 25 μL of serum ¼ diluted in PBS 1× FBS 1%. Reactivity of patient's sera for ECs was revealed by incubation with an FITC-conjugated F(ab')2 anti-human IgG (clone 30242; Bio rad, Hercule, CA, USA) for 20 min at 4 °C. The fluorescence level was expressed as mean fluorescence intensity. A serum containing an anti-HLA class I antibody directed against HLA typing of the endothelial cell lines was used as positive control. Negative controls were performed using a pool of human AB sera from healthy male donors.

Results are expressed as a ratio of the MFI obtained with patients' sera to that obtained with the negative control. A value of the ratio greater than 1.5 was considered as positive.

**HLA and KIR genotyping**. Donor and recipient HLA typing were performed by PCR-SSO reverse (One Lambda, Canoga Park, CA, USA). HLA-C1 and C2 groups were determined for the donors and recipients considering the HLA C typing obtained by PCR-SSO reverse (One Lambda). The presence or absence of Bw4 motif was determined for the donors and recipients, considering the HLA A and B typing obtained by PCR-SSO reverse (One Lambda).

Recipients were genotyped for the 14 KIR genes (2DL1, 2DL2, 2DL3, 2DL4, 2DL5, 2DS1, 2DS2, 2DS3, 2DS4, 2DS5, 3DL1, 3DL2, 3DL3, 3DS1) and two pseudogenes (2DP1, 3DP1) by PCR-SSO reverse (KIR SSO Genotyping Test, One Lambda and Lifecodes KIR Genotyping, Immucor).

**Genetic prediction of missing self**. 2DL1, 2DL2, 2DL3, 3DL1, 3DL2 inhibitory KIRs educated NK cells only when the recipient expressed their respective HLA class I ligand: KIR2DL1/C2; KIR2DL2/C1; KIR2DL3/C1; KIR3DL1/Bw4 and KIR3DL2/A*03, *11.

Genetic prediction of missing self was defined as the lack of expression by the graft of the type of HLA class I molecule able to bind to an educating inhibitory KIR of the recipient (Fig. 2a).

**Cell preparation and cultures**. The human erythroleukemia cell line K562, which lacks expression of any MHC molecules, was kindly provided by I. Doxiadis, University of Leiden, Netherlands. It was cultured in RPMI-1640 (ThermoFisher Scientific, Courtaboeuf, France) complemented with foetal bovine serum (FBS) 10% (Dutscher, Brumath, France), L-glutamine 2 mM (ThermoFisher Scientific), penicillin 100 U/mL, streptomycin 100 μM and HEPES 25 mM (ThermoFisher Scientific) (hereafter referred to as "complete RPMI").

Primary human arterial ECs were isolated from organ donors (agreement PFS08-017 from the Agence de la Biomédecine, https://www.agence-biomedecine.fr) and prospectively stored in the DIVAT biobank (no. of biocollection #02G55). They were cultured in endothelial cell growth medium 2 (Promocell, Heidelberg, Germany) in flasks coated with fibronectin (Promocell) or gelatine 1% (Sigma, Saint Quentin-Fallavier, France) and used between passages 2 and 7. For some experiments, ECs were activated with recombinant human TNF-α (100 U/mL; Peprotech) for 48 h.

PBMCs were isolated from the blood of healthy volunteers by Ficoll gradient centrifugation (Eurobio, Courtaboeuf, France). PBMCs were cultured overnight at 37 °C in 5% $CO_2$ in complete RPMI supplemented with recombinant human IL-2 (R&Dsystems, Minneapolis, MN, USA) or were maintained at 4 °C in complete RPMI. NK cells were purified (>90%) from PBMCs by negative selection with magnetic enrichment kits (Stemcell, Grenoble, France).

**Flow cytometry**. For NK cell count, 200 μL of blood was incubated with anti-CD45 (clone 30-F11, 1/400; BioLegend, London, UK), anti-CD3 (clone SK7, 1/10; BD biosciences, Le Pont de Claix, France) and anti-CD56 (clone NCAM16.2, 1/10; BD biosciences) antibodies. The samples were then incubated with a Lysing Solution (BD biosciences) to eliminate the red blood cells. Lymphocyte count was performed with ABX Pentra 60C+ (Horiba, Irvine, CA, USA).

For KIR phenotyping, single-cell suspensions of human PBMCs were incubated with a fixable viability dye (ThermoFisher Scientific) for 20 min at 4 °C. After washing, the cells were incubated first with anti-CD19 (clone HIB19, 1/10, BD biosciences), anti-CD14 (clone M5E2, 1/10; BD biosciences), anti-CD3 (clone SK7, 1/10; BD biosciences), anti-CD56 (clone NCAM16.2, 1/10; BD biosciences), anti-KIR3DL1 (clone DX9, 1/25; BD Biosciences), anti-KIR2DL1/S5 (clone 143211, 1/10; R&Dsystems), and anti-KIR2DL3 (clone 180701,1/10; R&Dsystems) antibodies for 15 min at room temperature and then with anti-KIR2DL1/S1 (clone EB6B, 1/25; Beckman Coulter, Villepinte, France), anti-KIR2DL2-3/S2 (clone GL183, 1/25; Beckman Coulter), and anti-KIR3DL1-2 (clone REA168, 1/10; Miltenyi Biotec, Bergisch Gladbach, Germany) antibodies for an additional 15 min. The cells were then fixed with paraformaldehyde 2% (ThermoFisher Scientific) and the sample was stored at 4 °C until analysis.

All sample acquisitions were made on a LSR FORTESSA or a FACScanto II® flow cytometer (BD biosciences) and analyses were performed with FlowJo software version 10.0.8r1 (Tree Star Inc, Ashland, OR, USA).

**Imaging flow cytometry**. Purified human NK cells ($10^5$) were mixed with K562 cells at a ratio of 1:1 in V-bottomed 96-well plates, centrifuged at $100g$ for 1 min, and incubated 30 min, 1 h, 2 h or 3 h at 37 °C at 5% $CO_2$. Negative controls were NK cells cultured alone and positive controls were NK cells cultured with IL-15 (100 ng/mL; Peprotech).

At indicated time points, the cells were harvested, stained with a fixable viability dye (ThermoFisher Scientific) and then surface-stained with anti-CD3 (clone SK7,

1/10; BD Biosciences), and anti-CD56 (clone NCAM16.2, 1/10; BD Biosciences) antibodies. The cells were subsequently fixed, permeabilised (Lysefix/PermIII® fixation/permeabilisation kit; BD Biosciences) and stained with anti-phospho-S6 ribosomal protein Ser 235/236 (clone D57.2.2E, 1/50; Cell Signaling Technology, Leiden, The Netherlands) or anti-PAkt S473 (clone M89-61, 1/40; BD Biosciences) antibodies.

Sample acquisitions were made on an ImageStream X Mark II (Amnis-EMD Millipore, Darmstadt, Germany) with ×40 magnification and analysed with IDEAS software (v6.0).

**NK cell activation in vitro**. For analysis of missing self-NK activation, PBMCs were cultured overnight in RPMI supplemented with 500 IU/mL of recombinant human IL-2 (R&Dsystems). Purified NK cells ($10^5$ cells) were then mixed with ECs at a ratio of 1:1 in flat-bottomed 96-well plates, centrifuged at $100g$ for 1 min, and incubated at 37 °C at 5% $CO_2$. Anti-CD107a-FITC (clone H4A3, 5 μL; Thermo-Fisher Scientific) was added prior the start of the assay. One hour after the beginning of the co-culture, Golgi Stop (BD Biosciences) was added to each well.

After 4 h of co-culture, the cells were harvested and surface-stained with appropriate antibody combinations to identify KIR subsets. The cells were subsequently fixed and permeabilised (Cytofix/Cytoperm fixation/permeabilisation kit; BD Biosciences), stained with anti-MIP-1ß-V450 (clone D21-1351, 1/40; BD biosciences) antibodies and analysed by flow cytometry.

For analysis of IL15-induced mTORC1 activation in NK cells, PBMCs of 24 patients diagnosed with breast cancer were collected before and one month after the introduction of a mTOR inhibitor (everolimus). PBMCs were cultured for 1 h in complete RPMI. When indicated, 100 ng/mL of IL-15 was added to the cultures. After 1 h, the cells were harvested and surface-stained with appropriate antibody combinations: anti-CD7 (clone 8H8.1, 1/50; Beckman Coulter) and anti-CD3 (clone SK7, 1/10; BD Biosciences). The cells were subsequently fixed and permeabilised (Cytofix/Cytoperm fixation/permeabilisation kit; BD Biosciences), stained with anti-phospho-S6 ribosomal protein Ser 235/236 (clone D57.2.2E, 1/50; Cell Signaling Technology, Leiden, The Netherlands) antibody and analysed by flow cytometry.

**In vitro cytotoxicity assays**. For analysis of endothelial cell viability, PBMCs were cultured overnight in RPMI supplemented with 60 IU/mL of recombinant human IL-2 (R&D Systems). In each culture well, $10^4$ human primary ECs (either Bw4$^-$ or Bw4$^+$) were seeded. After 24 h, $10^5$ purified NK cells from KIR3DL1$^+$ or KIR3DL1$^-$ donors were added to the culture. When indicated, 0.5 μg of anti-KIR3DL1 blocking monoclonal antibody (clone DX9; BD Biosciences) or an isotype control was added to the cultures.

Endothelial cell viability was monitored every 5 min for 10 h by electrical impedance measurement with an xCELLigence RTCA SP instrument (ACEA Biosciences, San Diego, CA, USA). The cell indices (CI) were normalised to the reference value (measured just prior to adding NK cells to the culture). Endothelial cell viability in the experimental well was normalised over the control well.

For analysis of K562 viability, PBMCs collected from eight healthy volunteers were co-cultured with 2500 K562 cells transfected with NanoLuc® luciferase at different effector-to-target ratios. When indicated, 25 nM of mTOR inhibitor (rapamycin) was added to the cultures. After 6 h of co-culture, 50 μL of supernatant of each well was collected and Nano-Glo® Luciferase Substrate (Promega, Madison, WI, USA) was added. K562 cell viability was assessed by measurement of luminescence for each well with an Infinite® 200 PRO instrument (TECAN, Männedorf, Switzerland).

**Mice**. Wild-type C57BL/6 (H-2$^b$) mice aged 8–15 weeks were purchased from Charles River Laboratories (Saint-Germain sur l'Arbresle, France).

C57BL/6 mice in which ß2-microglobulin gene has been deleted (hereafter referred as ß2-microglobulin KO) lack MHC class I protein expression on the cell surface. These mice were kindly provided by Laurent Genestier (CRCL, UMR INSERM 1042 CNRS 5286 Centre Léon Bérard).

All mice were maintained in germ-free conditions in our animal facility: Plateau de Biologie Expérimentale de la Souris (http://www.sfr-biosciences.fr/plateformes/animal-sciences/AniRA-PBES; Lyon, France).

All studies and procedures were performed in accordance with EU guidelines and were approved by the local ethical committee for animal research (CECCAPP Lyon, registered by the French National Ethics Committee of Animal Experimentation under No.15, http://www.sfr-biosciences.fr/ethique/experimentation-animale/ceccapp).

**NK cell depletion in vivo**. Where indicated, in the heart transplant model, mice were given intraperitoneally 100 μg of anti-NK1.1 monoclonal antibody (clone PB136; BioXcell, West Lebanon, NH, USA) twice a week from day −7 to the end of the experiment. NK cell depletion was verified by flow cytometry by quantifying the number of circulating Nkp46+ cells.

**Heart transplantation model**. Cardiac allografts were transplanted into sub-cutaneous space of the right neck. Anastomoses were performed by connecting the ascending aorta of the graft end-to-end with the recipient's common carotid artery and by pulling the main pulmonary artery with the external jugular vein as described in ref. [13]

Where indicated, mice were injected intraperitoneally with 100 μg of poly(I:C) (polyinosinic–polycytidylic acid; Invivogen, Toulouse, France) at day 4.

When indicated, the heart graft was kept at 4 °C for 3 h before transplantation to induce ischaemia/reperfusion injuries.

Heart transplants were harvested 60 days after transplantation, fixed in 4% buffered formalin for 24 h and embedded in paraffin for haematoxylin and eosin stain and immunohistochemistry. The following primary antibodies were used: anti-mouse CD31 (clone SZ31; 1/50; Dianova, Hamburg, Germany), anti-mouse CD45 (clone 30-F11; 1/40; BD Biosciences), and anti-Nkp46 (kind gift from Innate Pharma, Marseille, France) to stain, respectively, the ECs, the hematopoietic cells and the NK cells. The sections were revealed by Vectastain ABC HRP Kit (Vector, Peterborough, UK). The amount of labelled cells was semi-quantitatively assessed as follows: 0 normal; 1+ minimal or rare foci; 2+ moderate or several foci; 3+ marked or multifocal or diffuse.

When indicated, mice were given intraperitoneal injections of a calcineurin inhibitor (cyclosporin, Sandimmum; Novartis, Rueil-Malmaison, France) 20 mg/kg/day or an mTOR inhibitor (rapamycin; Bio Basic, Amherst, NY, USA) 3 mg/kg/day from day −7 to the end of the experiment.

**Statistical analyses**. For each dataset, mean ± standard deviation was calculated. For graphical presentation of the same data sets, box plots were generated using Prism software (Version 6.01; GraphPad Software Inc., La Jolla, CA), which presents the entire dataset distribution. The centre line in the boxes shows the medians; the box limits indicate the 25th and 75th percentiles; the whiskers indicate the 10th and 90th percentiles.

Differences between the groups were evaluated by the Mann–Whitney test, Fisher's exact test, unpaired $t$-test, one-way ANOVA followed by a Tukey's post hoc test, or by two-way ANOVA followed by a Šidák's post hoc test, according to the size of the groups and the distribution of the variable. All the tests used were two-sided. The test used for comparison is indicated in the figure legends.

Renal graft survivals were compared using the log-rank test.

Quadratic discriminant analysis was conducted on the CAGI dataset using the JMP software (Version 14; SAS Institute Inc., Cary, NC, USA).

The differences between the groups were considered statistically significant for $p < 0.05$ and are reported with asterisks (*$p < 0.05$; **$p < 0.01$; ***$p < 0.001$; ****$p < 0.0001$).

**Reporting summary**. Further information on research design is available in the Nature Research Reporting Summary linked to this article.

## Data availability
The authors declare that all data supporting the findings of this study are available in the article and its Supplementary Information Files, or on request from the corresponding author. The source data underlying Fig. 3A, Fig. 4C, D, Fig. 7B, C, Fig. 8B, Supplementary Fig. 4C, Supplementary Fig. 7A, Supplementary Table 1 are provided as a source data file.

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

## Acknowledgements

We thank Christelle Forcet and Violaine Tribollet for help with Xcelligence experiments, and Marie-Claude Gagnieu and Laurent Genestier for fruitful scientific discussions. O.T. is supported by the Agence nationale pour la Recherche (ANR-16-CE17-0007-01), the Fondation pour la Recherche médicale (PME20180639518), and the Etablissement français du Sang. C.-C.C., E.M. and O.T. are supported by the Institut Hospitalo-Universitaire–Organ Protection and Replacement (IHU-OPeRa; ANR-10-IBHU-004). A.K. is supported by INSERM (poste accueil 2015/1239/BT), the Hospices Civils de Lyon and the Fondation du Rein. SFR Biosciences is supported by the Agence nationale pour la Recherche (ANR -11-EQPX-0035 PHENOCAN). E.M. and O.T. are members of the CENTAURE Transplantation Research Network. The authors thank Mathilde Koenig for her contribution to the design of the figure.

## Author contributions

A.K., C.-C.C. and O.T. conceived and designed the experiments. A.K., C.-C.C., V.M., A.S., M. Racapé, S. Ducreux and V.-M.Y. performed the experiments. A.K., C.-C.C., T.B., V.M., M. Rabeyrin, P.B., J.-P.D.-V.-H., V.D., S. Dussurgey, V.M.-Y. and O.T. analysed the data. A.K. and O.T. wrote the paper; A.M., A.S., A.L., J.-C.O.-M., H.P., R.G., J.-L.T., J.C., E.M., A.N., B.C., M.N., T.W. and T.D. contributed to the discussion.

## Competing interests

The authors declare no competing interests.
