## [Peer Review File · Nature Communications]

Reviewers' comments:

Reviewer #1 (Remarks to the Author):

In this manuscript, König et al. analyse in detail the role of NK cells in chronic vascular rejection of solid organ transplantation. Starting from histologic analyses of microvascular inflammation detected in kidney transplantation cohort in absence of donor specific antibodies (DSA), the authors postulate a potential role of NK cells regarding the genetic of the KIR receptors content of the recipient, and the HLA ligand of the donor.

They came to the conclusion that, in the context of educated NK cells, the missing-self situation (absence of HLA ligand for inhibitory KIR) is associated with a higher prevalence of microvascular inflammation (without presence of DSA). By first in vitro experiments then in vivo mouse model of heart transplantation, they demonstrate the role of NK cells in the endothelial injury and inflammation which seems to be driven by the activation of mTOR pathway. Inhibition of mTOR pathway with rapamycin blocks this process.

They conclude that microvascular inflammation in absence of donor specific antibody is mediated by a direct killing of endothelial cells by NK cells in the context of missing-self situation.

The data is very robust, starting from clinical observation to in vitro model, then to a rodent model, to prove their hypothesis. The paper is of great quality, but several points should be considered to improve the quality of the manuscript.

- In Figure 1a, the groups of MVI with and without anti-HLA antibody seem to be comparable. This is a high number of MVI+/DSA- which could have been over estimated by the exclusion of anti-HLA antibodies that cannot be detected by the solid phase assay due false negative results. Have interference effects been excluded by treating the serum with EDTA, heat inactivated or other protocols?

- In Figure 1d the detection of NK cells in kidney biopsies is performed only with the CD56 which could be expressed by other cell types such as T-lymphocytes and monocytes. Is there any additional analysis performed to exclude these different cell types?

- In Figure 2a, the number of patients that are finally analysed and compared regarding missing-self are quite low. Is there any possibility for the author to include a validation cohort to reinforce their hypothesis based on genetic association?

- Figures 3 and 4 describe the functional effect of NK cells (CD107 and MIP-B expression) on target cells viability (endothelial cells) in the context of missing-self or non-missing-self situation. The functional effect of NK cells is mediated by absence of inhibition (KIR other other inhibitory receptors) but also by the activation through activating receptors. Is there any data showing which activating receptors are important to reduce the endothelial cells viability?

- The role of mTOR inhibition to block NK cells activity has already been published and references should be updated (Jensen H J Immunol 2017, Viel S Sci Signal 2016)

Reviewer #2 (Remarks to the Author):

Koenig et al. reported that in a cohort of 129 renal transplant patients, microvascular inflammation in graft biopsy was not mediated by antibodies in almost half of the cases. Recipients with typical AMR (group MVI+DSA+C3d+) had the worst graft survival compared to other 3 groups. Patients with antibody-independent microvascular inflammation (group MVI+DSA-) had the same graft survival as

the patients with AMR, due to non-complement activating DSA (group MVI+DSA+C3d). Graft survivals of these two groups were significantly better than that of MVI+DSA+C3d+ patients but significantly worse than the one of a matched control cohort without MVI or DSA (group MVI-DSA-). Immunohistopathologic examination revealed that NK cells were present in the graft microcirculation of MVI+DSA+C3d- patients. The recipients with antibody-independent microvascular inflammation had statistically more genetically-predicted missing self (MS) than matched controls. However, 34% of MVI+DSA- patients had no genetically-predicted missing self, indicating that other molecular mechanisms can also induce antibody-independent microvascular inflammation. The authors later did experimental studies and primary allogeneic human endothelial cells were co-cultured with NK cells purified from the PBMCs of healthy volunteers. After priming with low dose IL2, NK cells that could specifically detect the absence of expression of a particular HLA class I molecule (MS group) expressed significantly higher levels of both CD107a and MIP-1 β as compared to NK cells that did not express the specific inhibitory KIR (no MS) or that expressed the appropriate inhibitory KIR but were not educated (uneduc MS). The authors later demonstrated that demonstrate that the signals generated by each educating inhibitory KIR expressed on the surface are integrated by NK cells and modulate missing self-induced activation. In mice models, heart grafts, harvested in wild type C57B/L6 (controls) or β 2-microglobulin KO mice, were transplanted to wild type C57B/L6 mice. Priming of the recipients' NK cells induced by mild ischemia/reperfusion injuries resulted in the appearance of microvascular inflammation. Graft biopsies from patients with missing self-induced NK cell-mediated rejection confirmed that the mTORC1 pathway was activated in NK cells adherent to graft microvasculature. suppress missing self-induced cytotoxicity of human NK cells was evaluated ex vivo. NK cells were purified from the circulation of 24 patients before and 1 month after introduction of a mTOR inhibitor and the level of phosphorylation of S6 Ribosomal Protein (S6RP, which is located downstream mTORC1) was measured by flow cytometry. Exposition to the drug in vivo, not only decreased the baseline level of phosphorylation of S6RP in NK cells, but also drastically reduced their response to the stimulation by IL-15. Calcineurin inhibitor-treated animals developed the same microvascular inflammation as untreated controls but recipient mice treated with a mTOR inhibitor showed significantly less endothelial turgidity and inflammatory effectors in heart graft capillaries. This is a novel and excellent study conducted by the authors. I have a few clinical questions:

1. The authors claimed at the Introduction that the presence of microvascular inflammation in graft biopsy is widely considered as the histological hallmark of AMR. However, previous study by Gupta et al. investigating molecular and clinical significance of MVI showed only 54% of the patients with MVI score > 1 had DSA (Kidney International 2016; 89: 217-225) similar to authors' finding. MVI is documented in early deceased-donor kidney biopsies due to ischemia/reperfusion injury, in acute tubular necrosis (ATN), glomerulonephritis (GN), and acute T cell mediated rejection (TCMR) (Gibson et al. Am J Transplant. 2008;8(4):819-25).
2. The authors documented NK cell infiltration on the graft. Did the authors study T cell infiltrates at the biopsies? (CD3, CD4, CD8+ T cells). This will give information regarding role of T cell mediated mechanisms in development of MVI.
3. The authors studied anti-endothelial cell antibodies and identified in 6 patients. Did they study other non-HLA antibodies such as angiotensin receptor antibodies or MICA or MICB?
4. Although the authors showed mTORC1 inhibition efficiently prevents the development of histological lesions due to missing self-induced NK cell activation in a murine experimental model of transplantation, I would be cautious to interpret in treatment of chronic AMR, due to potential of mTOR inhibitors causing podocyte injury and increase proteinuria where chronic AMR patients already have proteinuria

Reviewer #1

In this manuscript, König et al. analyse in detail the role of NK cells in chronic vascular rejection of solid organ transplantation. Starting from histologic analyses of microvascular inflammation detected in kidney transplantation cohort in absence of donor specific antibodies (DSA), the authors postulate a potential role of NK cells regarding the genetic of the KIR receptors content of the recipient, and the HLA ligand of the donor.

They came to the conclusion that, in the context of educated NK cells, the missing-self situation (absence of HLA ligand for inhibitory KIR) is associated with a higher prevalence of microvascular inflammation (without presence of DSA). By first in vitro experiments then in vivo mouse model of heart transplantation, they demonstrate the role of NK cells in the endothelial injury and inflammation which seems to be driven by the activation of mTOR pathway. Inhibition of mTOR pathway with rapamycin blocks this process.

They conclude that microvascular inflammation in absence of donor specific antibody is mediated by a direct killing of endothelial cells by NK cells in the context of missing-self situation.

The data is very robust, starting from clinical observation to in vitro model, then to a rodent model, to prove their hypothesis. The paper is of great quality, but several points should be considered to improve the quality of the manuscript.

We are thankful to the reviewer for his positive appreciation of our work.

In Figure 1a, the groups of MVI with and without anti-HLA antibody seem to be comparable. This is a high number of MVI+/DSA- which could have been over estimated by the exclusion of anti-HLA antibodies that cannot be detected by the solid phase assay due false negative results. Have interference effects been excluded by treating the serum with EDTA, heat inactivated or other protocols?

We fully agree that interfering substances, such as IgM and complement, can be responsible for false negative results when screening the sera of patients for donor-specific anti-HLA antibodies with single antigen beads (SAB) assay (*Bettinoti et al, Curr Opin Organ Transplant. 2016*). Because interfering substances dilute to undetectable levels sooner than the HLA antibody, **dilution of sera before testing is considered as good laboratory practice** (*Bettinoti et al, Curr Opin Organ Transplant. 2016*) **and was already systematically performed in our study.**

To further address the reviewer's comment, Dr Dubois (head of Lyon University Hospital Immunogenetic laboratory) **retested all the sera of the cohort that were found positive in detection assay but without DSA in SAB assay. Following reviewer's advice, pretreatment with EDTA** was used to eliminate interference due to the binding of complement proteins (*Schnaidt et al, Transplantation, 2011*). Of the 28 tests performed (12 for anti-HLA I and 16 for anti-HLA II), only 1 detected of a previously missed anti-A36 DSA (1/28, 3.5%; MFI = 4300). The C3d binding assay being negative, this patient has been reclassified in the group MVI+DSA+C3d- and all the calculations have been modified accordingly in the revised version of the MS.

Changes:

Supplementary methods page 18:

“Serum samples banked at the time of biopsy from patients with significant microvascular inflammation were diluted 1/5 in washing buffer (following manufacturer instructions) and tested for the presence of donor specific anti-HLA antibodies using Screening Flow Beads (LifeScreen, Class I and Class II ID®, Lifecodes, Immucor) and Single Antigen Flow Beads (LSA class I and class II®, Lifecodes, Immucor) in case of positivity or questionable result of the screening test. To rule out a false negative result due to interference with complement proteins, all sera positive in Screening Flow Beads but without identified DSA in Single Antigen Flow Beads were retested after a pretreatment with EDTA².”

- In Figure 1d the detection of NK cells in kidney biopsies is performed only with the CD56 which could be expressed by other cell types such as T-lymphocytes and monocytes. Is there any additional analysis performed to exclude these different cell types?

In order to precisely quantify the performances of CD56 staining to identify NK cells we performed a flow cytometry analysis on the PBMCs of 4 healthy volunteers. The results, showed **Figure I** (for reviewers only), confirmed i) that virtually all NK cells (expressing the specific marker Nkp46) are CD56+ (**Figure I, middle row**), and ii) that ~3/4 of CD56+ cells are NK cells (**Figure I, lower row**).

Our results therefore confirm that, despite an excellent sensitivity, CD56 staining, which is routinely performed in pathology laboratories across the world to identify NK cells in tissues (including in kidney allograft biopsies: *Hidalgo et al, Am J Transplant, 2010*), is not fully specific (1/4 of CD56+ cells in the biopsy are Nkp46-). **This information has been added to the revised version of the MS.**

Despite all our efforts, we have been unable to re-analyse the biopsies of the study with Nkp46 staining. First, because Nkp46 staining did not give satisfactory results on the

preliminary experiments we conducted (not shown), likely because of inadequate fixation and/or antigen retrieval procedures. Second, because we ran out of biopsy material for many patients of the cohort due to the preparation of the additional slides required for CAGI analysis. **Of note however, CAGI analysis confirmed that graft infiltration by CD3+ and CD68+ cells were similar in the biopsies of patients from MVI+DSA+C3d- and MVI+DSA- groups (please see new Supplementary Figure 2A).**

According to **Figure I**, the vast majority of CD56+Nkp46- cells are CD3 positive, suggesting that T cells are the main contaminating cells in the histological analysis. We believe that this does not challenge the conclusion of our study that missing self-induced activation of NK triggers kidney graft rejection.

Indeed, CD56+CD3+ “contaminating” cells could be either NKT cells or a particular subpopulation of CD8+T cells (*Seyda et al, Trend in Immuno, 2016*):

- NKT have an invariant TCR and no study available in the literature has reported that they can be activated by direct allorecognition. In the other hand, because NKT cells, in addition to a TCR, are also equipped with KIRs, they are likely sensitive to missing-self (*Vivier et al, Nat Rev Immunol, 2012*).

- Previous studies have reported that the lysis capacity of the subpopulation of CD8+ T cells that express CD56 is negatively regulated by functional inhibitory KIR (*Phillips, et al, Science, 1995*). In other words, that the activation of CD56+ CD8+ T cells in graft microcirculation depends on the existence of a missing self.

=> this data highlights the importance of missing self for the development of MVI+DSA-lesions.

In our murine model of heart transplantation: i) rejection was abrogated in animals treated with anti-NK1.1, a monoclonal antibody that specifically deplete NK cells (but not T cells), and ii) cyclosporine, which is very efficient to prevent T cell activation, did not show any impact on the development of the lesions. Finally, an independent group (*Graham et al, Am J Transplant, 2009*) also reported that coronary allograft vasculopathy can develop via an NK-cell-dependent pathway (in the absence of T-lymphocytes) in a murine heart transplantation model.

=> this data points toward the crucial role of NK cells for the development of MVI+DSA-lesions.

Changes:

new Supplementary Figure 2A

Supplementary methods page 18:

“Double stainings with anti CD34 (endothelial cells) and respectively one antibody among anti CD3 (T cells), anti CD20 (B cells), anti CD66b (granulocytes), anti CD68 (macrophages) and anti CD56 (NK cells) were performed by immunohistochemistry on paraffin embedded sections using an anti-human CD34 (clone QBEnd10, Dako, Les Ulis, France) and respectively anti-human CD3 (clone SK7, Becton Dickinson, Le Pont de Claix, France), anti-human CD20 (Clone L26, Dako), anti CD66b (clone G10F5, Becton Dickinson), anti-human CD68 (clone PGM1, Dako) and anti-human CD56 (clone CD564, produced by Novocastra and distributed by Leica Microsystems SAS, Nanterre, France). Of note, CD56 staining, although extremely sensitive (94%) is not fully specific for NK cells. About 3/4 of CD56+ cells are NK (Nkp46+). The majority of CD56+ Nkp46- cells are either NKT or CD8+T cells.”

- In Figure 2a, the number of patients that are finally analysed and compared regarding missing-self are quite low. Is there any possibility for the author to include a validation cohort to reinforce their hypothesis based on genetic association?

We agree with the reviewer that a validation cohort is a very relevant approach to i) confirm new clinical findings and ii) appreciate their importance in “real life”.

Our study enrolled all renal transplant patients followed in our center for whom a graft biopsy had been performed between 2004 and 2012.

A validation cohort made from patients of our center with a graft biopsy performed after 2012 would not be ideal because i) this would be an independent **but not an external** validation, ii) it would introduce a bias due to **significantly shorter follow-up time**.

Instead we have **confirmed the validity of our theory** by combining several independent *in vitro* and *in vivo* experimental approaches, which all gave concordant results.

The estimation of the **clinical importance of our theory** will require validation in an independent external cohort. We have taken preliminary contacts with other transplant centers, but this hard work will certainly be eased by the publication of the princeps manuscript.

- Figures 3 and 4 describe the functional effect of NK cells (CD107 and MIP-B expression) on target cells viability (endothelial cells) in the context of missing-self or non-missing-self situation. The functional effect of NK cells is mediated by absence of inhibition (KIR other inhibitory receptors) but also by the activation through activating receptors. Is there any data showing which activating receptors are important to reduce the endothelial cells viability?

The reviewer is right in pointing out the fact that NK cell activation is governed by the integration of activating and inhibitory signals. Recipient’s NK cells could therefore be activated by signaling through activating KIR instead of (or in addition to) a lack of signaling through inhibitory KIR.

KIR locus is highly polymorphic for allele and gene content (*Thielens A et al, Current opinion in immunology 2012*). At the population level, 2 major KIR haplotypes can be defined. Haplotypes A and B share inhibitory KIR content but differ strongly in their activating KIR content: Haplotype A patients have only one activating KIR (KIR2DS4) whereas those of haplotype B have multiple activating KIR (*Parham P et al, Seminars in immunology 2008*). In order to evaluate the importance of activating KIR in the generation of DSA-independent MVI lesions, we compared the number and distribution of activating KIR genes between MVI-DSA- (control group, n=55) and MVI+DSA- (n= 43) patients. No difference was found between the two groups, demonstrating that **patients that have more activating KIR genes do not have increased risk to develop DSA-independent MVI lesions**. These new data are presented in the **revised version of Supplementary Table 2**.

This genetic analysis does not rule out the involvement of activating KIR in DSA-independent MVI. NK cell activation indeed also depends upon the expression of the ligands by graft endothelium (as demonstrated for inhibitory KIRs). A major limitation to test this hypothesis is the fact that the ligands of activating KIR are not as well defined as those of inhibitory KIRs (*Ivarsson et al, Front Immunol, 2014*).

In vivo murine model of heart transplantation is not adapted to address this issue because cold ischemia, which is required to prime NK cells through liberation of DAMPS, also upregulate the expression of activating KIR ligands on stressed graft endothelial cells (*Raulet et al, Annu Rev Immunol, 2013*). Therefore, to evaluate the importance of activating KIRs in the process of NK cell activation by endothelial cells we used an *in vitro* approach. Human primary endothelial cells have been used in our study instead of an endothelial cell line because, due to its cancerous nature the latter can express activating KIR ligands at baseline (*Brandt et al, J Exp Med, 2009*). Primary endothelial cells have been cocultured with purified human NK cells that were primed or not with low dose IL2. Activation status of NK cells in coculture was evaluated by flow cytometry (CD107a and MIP1 β) and the viability of endothelial cells was monitored by real time measurement of the impedance. To test the importance of activating KIRs in the process of NK cell activation by endothelial cells, we have compared the results obtained when endothelial cells were previously exposed or not to TNF α , an inflammatory cytokine known to upregulate the expression of activating KIR ligands, including MIC (*Lin et al, JBC, 2012 & Chauveau, J Innate Immun, 2014*).

Using this model, we were able to demonstrate that **priming was necessary for NK cells to react to a lack of ligand of inhibitory KIRs (missing self)**. In contrast, **exposing endothelial cells to TNF α : i) was neither necessary nor sufficient to trigger NK cell activation per se** (even after NK cell priming with IL2; revised Supplementary Figure 4A), and **ii) neither increased the level of activation of primed NK cells (revised Supplementary Figure 4A), nor worsen the survival of endothelial cells in situation of missing self (new Supplementary Figure 5A)**.

Although this new set of data suggests that (in contrast to inhibitory KIRs) activating KIRs do not seem to play an important role in antibody-independent NK cell-mediated MVI development, it has been obtained in an artificial *in vitro* model. It is therefore conceivable that MVI+DSA- patients without genetically-predicted missing self (35% of MVI+DSA- patients, **Figure 2B**), have developed the lesions due to an excess of signaling through NK cells' activating KIRs. This possibility is discussed in the revised version of the MS.

Changes:

revised Supplementary Table 2

revised Supplementary Figure 3A

new Supplementary Figure 4

Main body page 12:

"Of note, ~1/3 (15/43; 34.9%) of MVI+DSA- patients had no genetically-predicted missing self, indicating that other molecular mechanisms can also induce antibody-independent microvascular inflammation (Figure 2B). NK cell activation is governed by the integration of the signals provided by inhibitory and activating KIRs³⁷. It is therefore tempting to speculate that in some patients, the activation of recipients' NK cells by graft endothelium was triggered by signaling through activating KIR instead of (or in addition to) a lack of signaling

*through inhibitory KIRs. However, we were unable to confirm this hypothesis since: i) the number and distribution of activating KIR genes were similar between MVI-DSA- and MVI+DSA- patients (**Supplementary Table 2**), and ii) increasing the expression of KIR activating ligands on endothelial cells was neither necessary nor sufficient to trigger NK cell activation and to promote endothelial cells damages (**Supplementary Figure 4A and Supplementary Figure 5A**)."*

- The role of mTOR inhibition to block NK cells activity has already been published and references should be updated (Jensen H J Immunol 2017, Viel S Sci Signal 2016)

We thank the reviewer for pointing these two references to us. They are now cited in the revised MS.

Changes:

See references # 49 and 50 in the bibliography section of the revised MS

Reviewer #2

Koenig et al. reported that in a cohort of 129 renal transplant patients, microvascular inflammation in graft biopsy was not mediated by antibodies in almost half of the cases. Recipients with typical AMR (group MVI+DSA+C3d+) had the worst graft survival compared to other 3 groups. Patients with antibody-independent microvascular inflammation (group MVI+DSA-) had the same graft survival as the patients with AMR, due to non-complement activating DSA (group MVI+DSA+C3d). Graft survivals of these two groups were significantly better than that of MVI+DSA+C3d+ patients but significantly worse than the one of a matched control cohort without MVI or DSA (group MVI+DSA-). Immunohistopathologic examination revealed that NK cells were present in the graft microcirculation of MVI+DSA+C3d- patients. The recipients with antibody-independent microvascular inflammation had statistically more genetically-predicted missing self (MS) than matched controls. However, 34% of MVI+DSA- patients had no genetically-predicted missing self, indicating that other molecular mechanisms can also induce antibody-independent microvascular inflammation. The authors later did experimental studies and primary allogeneic human endothelial cells were co-cultured with NK cells purified from the PBMCs of healthy volunteers. After priming with low dose IL2, NK cells that could specifically detect the absence of expression of a particular HLA class I molecule (MS group) expressed significantly higher levels of both CD107a and MIP-1 β as compared to NK cells that did not express the specific inhibitory KIR (no MS) or that expressed the appropriate inhibitory KIR but were not educated (uneduc MS). The authors later demonstrated that demonstrate that the signals generated by each educating inhibitory KIR expressed on the surface are integrated by NK cells and modulate missing self-induced activation. In mice models, heart grafts, harvested in wild type C57B/L6 (controls) or β 2-microglobulin KO mice, were transplanted to wild type C57B/L6 mice. Priming of the recipients' NK cells induced by mild ischemia/reperfusion injuries resulted in the appearance of microvascular inflammation. Graft biopsies from patients with missing self-induced NK cell-mediated rejection confirmed that the mTORC1 pathway was activated in NK cells adherent to graft microvasculature. suppress missing self-induced cytotoxicity of human NK cells was evaluated ex vivo. NK cells were purified from the circulation of 24 patients before and 1 month after introduction of a mTOR inhibitor and the level of phosphorylation of S6 Ribosomal Protein (S6RP, which is located downstream mTORC1) was measured by flow cytometry. Exposition to the drug in vivo, not only decreased the baseline level of phosphorylation of S6RP in NK cells, but also drastically reduced their response to the stimulation by IL-15. Calcineurin inhibitor-treated animals developed the same microvascular inflammation as untreated controls but recipient mice treated with a mTOR inhibitor showed significantly less endothelial turgidity and inflammatory effectors in heart graft capillaries.

This is a novel and excellent study conducted by the authors. I have a few clinical questions:

We thank the reviewer for his encouraging comment.

1. The authors claimed at the Introduction that the presence of microvascular inflammation in graft biopsy is widely considered as the histological hallmark of AMR. However, previous study by Gupta et al. investigating molecular and clinical significance of MVI showed only 54% of the patients with MVI score > 1 had DSA (Kidney International 2016; 89: 217-225)

similar to authors' finding. MVI is documented in early deceased-donor kidney biopsies due to ischemia/reperfusion injury, in acute tubular necrosis (ATN), glomerulonephritis (GN), and acute T cell mediated rejection (TCMR) (Gibson et al. *Am J Transplant*. 2008;8(4):819-25).

These two references are congruent with our findings and both have been added to the bibliography of the revised MS.

Changes:

See references # 20 and 21 in the bibliography section of the revised MS

2. The authors documented NK cell infiltration on the graft. Did the authors study T cell infiltrates at the biopsies? (CD3, CD4, CD8+ T cells). This will give information regarding role of T cell mediated mechanisms in development of MVI.

We therefore thank the reviewer for his constructive suggestion. In order to address his comment, the nature and number of immune cells infiltrating the renal allograft of patients with available biopsy material from MVI+DSA+C3d+ (n=17), MVI+DSA+C3d- (n=14), and MVI+DSA- (n=32) groups were compared using the Computer-assisted Analysis of Graft Inflammation (CAGI) method that we recently published (*Sicard et al, Kidney int, 2017*). This approach allows for a precise quantification of the innate (CD68+ macrophages, CD66b+ neutrophils, and CD56+ NK cells) and the adaptive (CD3+ T cells, and CD20+ B cells) immune cells in the microcirculation (glomeruli and peritubular capillaries) and the tubulo-interstitial compartment of renal allograft.

Data of individual cell subsets did not allow discriminating the 3 groups of patients. In particular, there was no significant difference in CD3+ cells density between the 3 groups in neither of the 2 compartments of renal allograft analysed (**Supplementary Figure 2A**).

Quadratic discriminant analysis conducted on the whole dataset correctly separated MVI+DSA+C3d+ and MVI+DSA- patients but a major overlap was observed between MVI+DSA+C3d- and MVI+DSA-, which suggests that a common pathophysiological process is taking place in the renal allografts of these two groups (**Figure 1D**).

Since several previously published studies have documented the crucial role of NK cells in complement-independent AMR (*Hirohashi et al, Am. J. Transplant, 2012 & Hidalgo et al, Am. J. Transplant 2010*), we have compared the number of NK cells in the biopsy of MVI+DSA+C3d- and MVI+DSA- patients. Interestingly, NK cell density was similar in the two groups, indicating that a final common pathway involving NK cells could be either triggered classically by the humoral arm of the adaptive immune system of the recipients or induced by a direct antibody-independent activation of innate effectors.

Changes:

new Supplementary Figure 2A

revised Figure 1

Main body page 10:

"The nature and number of immune cells infiltrating the renal allograft of patients with available biopsy material from MVI+DSA+C3d+ (n=17), MVI+DSA+C3d- (n=14), and MVI+DSA-

(n=32) groups were compared using the Computer-assisted Analysis of Graft Inflammation (CAGI) method³². This approach allows for a precise quantification of the innate and adaptive immune cell subsets density in the microcirculation (within glomerular- and peritubular-capillaries) and the tubulo-interstitial compartment of renal allograft (**Supplementary Figure 2A & 2B**). The quadratic discriminant analysis conducted on CAGI dataset efficiently separated MVI+DSA+C3d+ and MVI+DSA- patients but a major overlap was observed between MVI+DSA+C3d- and MVI+DSA- (**Figure 1D**), which suggests that a common pathophysiological process is taking place in the renal allografts of these two groups.

Antibodies that are unable to activate the complement cascade can still recruit Fcγ receptor-expressing innate immune effectors, which are responsible for antibody-dependent cell mediated cytotoxicity (ADCC), thus leading to chronic humoral rejection⁷. Seminal experimental studies^{18,33}, confirmed by subsequent clinical observations³⁴, have demonstrated that among innate immune effectors subset, NK cells are crucial for the development of complement-independent humoral rejection lesions. In line with this data, NK cells were present in the graft microcirculation of MVI+DSA+C3d- patients (**Figure 1E**). Interestingly, NK cell infiltration was similar in MVI+DSA- patients, whose microvascular inflammation was not triggered by antibody deposition on the graft endothelium (**Supplementary Figure 2B**). We therefore concluded that in chronic vascular rejection, a final common pathway involving NK cells could be either triggered classically by the humoral arm of the adaptive immune system of the recipients or induced by a direct antibody-independent activation of innate effectors.”

3. The authors studied anti-endothelial cell antibodies and identified in 6 patients. Did they study other non-HLA antibodies such as angiotensin receptor antibodies or MICA or MICB?

Several independent studies have reported a possible deleterious impact of antibodies non directed against canonical HLA molecules (HLA A, B, C, DR, DQ, or DP): i.e. antibodies present in the circulation of recipients that are directed against an auto- or minor allo-antigen (*Delville et al, JASN, 2019 & Dragun et al, N Engl J Med, 2005 & Zou et al, N Engl J Med, 2007*).

The main problem with non-HLA antibodies is that their antigenic targets have not been fully identified yet (and that the list of the possible targets is expanding fast with the refinement of the techniques used to screen the serum of transplant patients). **Screening for all the possible specificities of non-HLA antibodies is therefore impossible.**

Importantly, the pathogenicity of non-HLA antibodies has not been tested for all the identified specificities. Based on what is known of the immunopathology of anti-HLA antibody-mediated rejection (in particular the fact that the toxicity of anti-HLA DSA is limited to graft microvasculature because IgG are sequestered in the circulation due to their size), it is likely that only antibodies directed against proteins expressed on the surface of graft endothelial cells can have a deleterious impact (*Chen et al, J Clin Invest, 2018*). This is the reason why we initially decided to screen for non-HLA antibodies with the endothelial flow cross match assay. **Our results suggest that non-HLA antibody-mediated rejections exist but are rare.** Of note, the low incidence of non-HLA antibodies in our cohort (6/129, 4,6%) could be due to the lack of sensitivity of the assay. However, this limit does not challenge

our conclusions since this problem would only result in **contaminating the group of MVI+DSA- with “genuine” antibody-mediated rejection, thereby reducing our chance to detect the increase proportion of patients with mismatches between donor HLA class I and recipient inhibitory KIR receptors.**

Among the few non-HLA antibodies for which the pathogenicity has been demonstrated are anti-angiotensin II type 1–receptor (AT1R; *Dragun et al, N Engl J Med, 2005*) and anti-MHC class I polypeptide-related sequence (MIC; *Zou et al, N Engl J Med, 2007*). In order to address the reviewer query, the sera of all the patients enrolled in our study (MVI+DSA-; MVI+DSA-, MVI+DSA+C3d-, and MVI+ DSA+C3d+) were tested for the presence of anti-AT1R and anti-MIC antibodies using a highly sensitive Luminex assay. The results (presented in **revised Supplementary Figure 1A**), show that neither the MFI nor the proportion of positive patients are different between the 4 groups for these two non-HLA antibodies.

Changes:

revised Supplementary Figure 1A

Main body page 8:

*“Previous studies have shown that bona fide humoral rejections can be triggered by non-HLA antibodies directed against either minor histocompatibility alloantigens or autoantigens^{1,23–27}. Screening of patients’ sera for anti-angiotensin II type 1-receptor (AT1R) and anti-MHC class I polypeptide-related sequence (MIC), two types of non-HLA antibodies for which pathogenicity has been previously demonstrated^{28,29}, showed that neither the titres nor the proportion of positive patients were increased in the MVI+DSA- group (**Supplementary Figure 1A**). Although it is impossible to test for all non-HLA specificities, previous works have demonstrated that only antibodies able to bind to the surface of graft endothelial cells can have a deleterious impact¹⁴. Flow cytometric crossmatch with activated HLA-matched endothelial cells³⁰ was therefore used to screen the sera of the patients for non-HLA anti-endothelial cell antibodies (**Supplementary Figure 1B**). These experiments identified 6 patients (6/129, 4,6%) for which non-HLA anti-endothelial cell antibodies could account for graft microvascular inflammation (**Supplementary Figure 1C, 1D & Figure 1A**). Based on these results we concluded that in almost half of the cases (53/129, 41,1%; group MVI+DSA-), graft microvascular inflammation was not caused by host humoral response.”*

4. Although the authors showed mTORC1 inhibition efficiently prevents the development of histological lesions due to missing self-induced NK cell activation in a murine experimental model of transplantation, I would be cautious to interpret in treatment of chronic AMR, due to potential of mTOR inhibitors causing podocyte injury and increase proteinuria where chronic AMR patients already have proteinuria.

We fully agree with the reviewer’s comment. Missing-self induced NK-cell activation promote the development of microvascular lesions in the graft, including glomerulitis. Adaptation to stress by podocytes require signaling through mTOR pathways (*Yao et al, Curr Opin Nephrol Hypertens, 2016 & Canaud et al, Nat Med, 2013*). It is therefore likely that if introduced late in the course of the disease, mTOR inhibitors will worsen glomerular lesions. This element of discussion has been added to the revised version of the MS.

Changes:

Main body page 23:

“A note of caution shall however be sounded in the case of renal transplantation. NK cell-induced microvascular lesions in the glomeruli trigger significant stress on podocytes. These key cellular players of the filtration barrier are known to adapt to stress through mTOR pathways^{59,60} and it is therefore possible that introduction of mTOR inhibitors late in the course of the disease (i.e. in presence of proteinuria and/or chronic glomerular lesions on the biopsy) could be ill tolerated in these patients.”

REVIEWERS' COMMENTS:

Reviewer #1 (Remarks to the Author):

The author responds adequately to most of the reviewer comments. One point should be clarified regarding activating receptors. In their answer on activating receptors, the author focused on activating KIRs. But other activating receptors are important such as NKG2D,NKG2C, NKp46,NKp30... Is there any data showing which activating receptors (listed above) are important to reduce the endothelial cells viability?. This should be at least include in the discussion, because this are potential targets for therapy.

Reviewer #2 (Remarks to the Author):

I accept the revised manuscript for publication

POINT-BY-POINT RESPONSE TO REVIEWERS' COMMENTS:

We are thankful to the Editor and the 2 Reviewers for their critics and suggestions, which have significantly contributed to increase the quality of our study.

Reviewer #1 (Remarks to the Author):

The author responds adequately to most of the reviewer comments. One point should be clarified regarding activating receptors. In their answer on activating receptors, the author focused on activating KIRs. But other activating receptors are important such as NKG2D, NKG2C, NKp46, NKp30... Is there any data showing which activating receptors (listed above) are important to reduce the endothelial cells viability?. This should be at least include in the discussion, because this are potential targets for therapy.

We fully agree that, beyond KIR, many other activating receptors could theoretically promote the activation of recipient's NK cells (and therefore trigger NK cell-mediated rejection). Following the reviewer's advice, a sentence has been added to the revised version of the manuscript to clearly indicate this possibility.

Changes:

Main body, page 11:

"These results however do not rule out the role of activating receptors in the triggering of NK cell-mediated rejection. Indeed, beyond activating KIRs, many other type of activating receptors exist on NK cells (including, NKG2D, NKG2C, NKp46, NKp30...etc)."